# FEATURE HEDGING: CORRELATED FEATURES BREAK NARROW SPARSE AUTOENCODERS

## ABSTRACT

It is assumed that sparse autoencoders (SAEs) decompose polysemantic activations into interpretable linear directions, as long as the activations are composed of sparse linear combinations of underlying features. However, we find that if an SAE is more narrow than the number of underlying "true features" on which it is trained, and there is correlation between features, the SAE will merge components of correlated features together, thus destroying monosemanticity. In LLM SAEs, these two conditions are almost certainly true. This phenomenon, which we call *feature hedging*, is caused by the SAE's reconstruction loss, and is more severe the narrower the SAE. In this work, we introduce the problem of feature hedging and study it both theoretically in toy models and empirically in SAEs trained on LLMs. We suspect that feature hedging may be one of the core reasons that SAEs consistently underperform supervised baselines. Finally, we use our understanding of feature hedging to propose an improved variant of matryoshka SAEs. Importantly, our work shows that SAE width is not a neutral hyperparameter: narrower SAEs suffer more from hedging than wider SAEs.

## 1 INTRODUCTION

As large language models (LLMs) are deployed in real-world applications, it is increasingly important to understand their internal workings. Sparse autoencoders (SAEs) decompose the dense, polysemantic activations of LLMs into interpretable latent features (Cunningham et al., 2024; Bricken et al., 2023) using sparse dictionary learning (Olshausen & Field, 1997). SAEs have the advantage of operating completely unsupervised, and can easily be scaled to millions of neurons in its hidden layer (hereafter called "latents" [1])(Templeton et al., 2024; Gao et al., 2024).

While SAEs showed promising results, recent work has cast doubt on the performance of SAEs relative to baseline techniques. Wu et al. (2025) show that SAEs underperform on both concept steering and detection relative to baselines, and Kantamneni et al. (2025) show that SAEs underperform simple linear probes on both in-domain and out-of-domain detection, even when the probes have very few training samples. The question, then, is why do SAEs underperform relative to other techniques? And if we can identify the problems holding back SAEs, can we then fix those problems?

One fundamental issue with SAEs is the problem of feature absorption (Chanin et al., 2024), where a more specific latent suppresses the firing a more general latent. For instance, an SAE may have a latent that appears to track "Cities in USA" but that arbitrarily fails to fire on the specific cities "New York" and "Detroit", where a city-specific latent fires instead. Feature absorption requires underlying features to exist in a hierarchy, with a parent feature $f_p$ and a child feature $f_c$, where $f_c$ can only fire if $f_p$ is firing ($f_c \implies f_p$). Feature absorption is caused by SAE sparsity penalty, and becomes more severe the wider the SAE. An SAE encoder/decoder under feature absorption is shown in Figure 1b.

In this paper, we identify another fundamental issue with SAEs which we call feature hedging. In hedging, an SAE is too narrow to represent both features $f_a$ and $f_b$ with their own latents $l_a$ and $l_b$. Ideally, an SAE should assign a latent $l$ to either $f_a$ or $f_b$, and ignore the feature not being tracked. However, if $f_a$ and $f_b$ are either hierarchical as in absorption, or (anti-)correlated, then the SAE

---

[1]We use the term "latents" for the hidden neurons of the SAE to avoid overloading the term "feature". We use "feature" only to describe interpretable concepts represented by the model.

Table 1: Comparing feature hedging and feature absorption

| Feature absorption | Feature hedging |
| --- | --- |
| Learns gerrymandered latents | Learns polysemantic mixtures of features |
| Caused by sparsity loss | Caused by MSE reconstruction loss |
| Features are all tracked in the SAE | One feature is in the SAE, the other is not |
| Affects the encoder and decoder asymmetrically | Affects encoder and decoder symmetrically |
| Gets worse the wider the SAE | Gets worse the narrower the SAE |
| Requires hierarchical features | Requires only correlation between features |

latent $l$ can reduce reconstruction error by incorrectly mixing in components of both $f_a$ and $f_b$. A sample SAE encoder and decoder experiencing hedging is shown in Figure 1a. In an LLM SAE, hedging will look like each SAE latent has noise mixed into it, reducing the performance of the latent for both detection and steering. Unlike with absorption, hedging becomes worse the narrower the SAE: thus trying to reduce absorption by making the SAE narrower will simply result in more hedging instead. The differences between hedging and absorption are shown in Table 1.

In LLM SAEs, the SAE is almost certainly narrower than the number of underlying features, as even extremely wide LLM SAEs appear to miss features (Templeton et al., 2024). Furthermore, we expect that nearly every feature in an LLM has positive and negative correlations to many features. We thus expect that hedging is the norm in LLM SAEs and will significantly distort their performance.

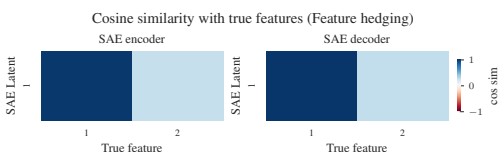

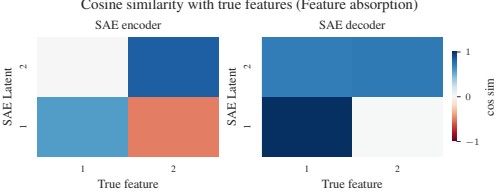

(a) When the SAE is only wide enough to represent one of the two features, we see feature hedging. Latent $l_1$ mainly tracks $f_1$, but a small component of $f_2$ is incorrectly mixed into the latent $l_1$ as well. $f_2$ is mixed symmetrically into both the encoder and decoder.

(b) Adding a new latent to the SAE so it is wide enough to track both features, we see feature absorption. The decoder for $l_1$ perfectly tracks $f_1$, but its encoder turns off if $f_2$ is also active. $l_2$ tracks $f_2$, but its decoder mixes $f_1$ and $f_2$. Asymmetry between encoder and decoder is characteristic of absorption.

Figure 1: SAE encoder and decoder patterns for hierarchical features $f_1$ and $f_2$, where $f_1 \implies f_2$. These features lead to either hedging or absorption depending on the width of the SAE.

A solution to feature absorption has been proposed in the form of matryoshka SAEs (Bussmann et al., 2025). Matryoshka SAEs use nested SAE loss terms to enforce a hierarchy on the SAE latents, solving absorption by forcing the narrow inner levels of the SAE to reconstruct inputs on their own. However, as we show in this paper, matryoshka SAEs suffer more from hedging due to the inner matryoshka levels essentially being very narrow SAEs. Matryoshka SAEs thus trade off absorption for hedging.

In this work, we define and study feature hedging both theoretically in toy models and empirically in LLM SAEs. We show that hedging is worse the more narrow the SAE, and introduce a technique to characterize the amount of hedging present in a given SAE. We also study hedging and absorption in matryoshka SAEs, and show that it is possible to improve the monosemanticity of matryoshka SAEs by tuning the relative loss coefficients in each level of the matryoshka SAE to better balance the competing forces of absorption and hedging—though both problems remain present. We show as well that SAE width is not a neutral hyperparameter: narrow SAEs suffer more from hedging than wider SAEs.

## 2 BACKGROUND

**Sparse autoencoders (SAEs).** An SAE decomposes an input activation $x \in \mathbb{R}^D$ into a hidden state $f$ consisting of $L$ hidden neurons, called "latents". An SAE is composed of an encoder $W_{\text{enc}} \in \mathbb{R}^{L \times D}$, a decoder $W_{\text{dec}} \in \mathbb{R}^{D \times L}$, a decoder bias $b_{\text{dec}} \in \mathbb{R}^D$, and encoder bias $b_{\text{enc}} \in \mathbb{R}^L$, and a nonlinearity $\sigma$, typically ReLU or a variant like JumpReLU (Rajamanoharan et al., 2024), TopK (Gao et al., 2024) or BatchTopK (Bussmann et al., 2024).

$$z = \sigma(W_{\text{enc}}(x - b_{\text{dec}}) + b_{\text{enc}}) \tag{1}$$
$$\hat{x} = W_{\text{dec}}z + b_{\text{dec}} \tag{2}$$

The SAE is trained with a reconstruction loss, typically Mean Squared Error (MSE), and a sparsity-inducing loss consisting of a function $\mathcal{S}$ that penalizes non-sparse representation with corresponding sparsity coefficient $\lambda$. For standard L1 SAEs, $\mathcal{S}$ is the L1 norm of $f$. For TopK and BatchTopK SAEs, there is no sparsity-inducing loss ($\mathcal{S} = 0$) as the TopK function directly induces sparsity. There is sometimes also an additional auxiliary loss $\mathcal{L}_{aux}$ with coefficient $\alpha$ to ensure all latents fire. Standard L1 SAEs typically do not have an auxiliary loss (Olah et al., 2024). The general SAE loss is

$$\mathcal{L} = \|x - \hat{x}\|_2^2 + \lambda\mathcal{S} + \alpha\mathcal{L}_{\text{aux}}. \tag{3}$$

**Tied SAEs.** A tied SAE has $W_{\text{enc}} = W_{\text{dec}}^\mathsf{T}$. The biases have different dimensions and are untied.

**Matryoshka SAEs.** A matryoshka SAE (Bussmann et al., 2025) extends the SAE definition by summing losses created by prefixes of SAE latents. This forces each sub-SAE to reconstruct input activations on its own, and incentivizes the SAE to place more common, general concepts into latents with smaller index number. A matryoshka SAE uses nested prefixes with sizes $\mathcal{M} = m_1, m_2, ...m_n$ where $m_1 < m_2 < \ldots < m_n = L$, where $L$ is the number of latents in the full dictionary. Matryoshka SAE loss is:

$$\mathcal{L} = \sum_{m \in \mathcal{M}} \left( \|x - \hat{x}_m\|_2^2 + \lambda\mathcal{S}_m \right) + \alpha\mathcal{L}_{\text{aux}} \tag{4}$$

Where $\hat{x}_m$ is the reconstruction for the SAE using the first $m$ latents, and $\mathcal{S}_m$ is the sparsity penalty applied to the first $m$ latents. For TopK and BatchTopK Matryoshka SAEs, there is no sparsity penalty ($\mathcal{S}_m = 0$) as the TopK function directly imposes sparsity.

## 3 TOY MODELS OF FEATURE HEDGING

The linear representation hypothesis (LRH) states that features in LLMs are represented as nearly-orthogonal linear directions in representation space (Bricken et al., 2023). The goal of SAEs, then, is to recover these underlying "true features" of the model, where each latent of the SAE decoder perfectly matches an underlying feature of the model. In real LLMs we do not have ground-truth knowledge of these underlying features, making it difficult to know if SAEs are succeeding at recovering model features. Fortunately, it is easy to create synthetic training data for SAEs that follows the LRH and gives us ground-truth knowledge of the underlying features. This allows us to understand when SAEs will learn the underlying features of the model, and when SAEs fail.

We define a toy model consisting of $N$ *true features* $F \in \mathbb{R}^{N \times D}$, where each $\|f_i\|_2 = 1$. These features are mutually orthogonal, so $\forall i \neq j, f_i \cdot f_j = 0$. Each feature $f_i$ has a corresponding firing probability $p_i \in [0, 1]$. For each sample, we generate a binary activation vector $\mathbf{a} \in \{0, 1\}^N$ where $a_i \sim \text{Bernoulli}(p_i)$ indicates whether feature $f_i$ is active (fires). The model can incorporate feature dependencies by conditioning firing probabilities: $a_i | \mathbf{a}_{-i} \sim \text{Bernoulli}(p_i(\mathbf{a}_{-i}))$, where $\mathbf{a}_{-i}$ denotes the activation states of all other features. We then generate SAE training samples $x$ from this model as $x = \sum_{i=1}^N a_i f_i$.

We say that an SAE is *correct* or *monosemantic* for this toy model if every latent in the SAE dictionary matches a true feature direction, and each SAE latent corresponds to a different true feature. Formally, there exists a bijection $\pi : \{1, \ldots, L\} \to \{1, \ldots, N\}$ such that $\cos(W_{\text{dec},i}, f_{\pi(i)}) = 1$ for all $i \in \{1, \ldots, L\}$. We only investigate SAEs where $L \leq N$ in our toy experiments. We say an SAE is *polysemantic* if some SAE latents contain positive or negative components of multiple true features, so there exists at least one latent $i \in \{1, \ldots, L\}$ such that $|\{j \in \{1, \ldots, N\} : |W_{\text{dec},i} \cdot f_j| > \epsilon\}| > 1$ for some threshold $\epsilon > 0$.

For all SAEs in this section, we train on 15M synthetic activations using SAELens (Bloom et al., 2024). In this section we show plots of the cosine similarity between the SAE encoder / decoder and the true features. Each cell $i, j$ in these plots is simply $\cos(W_{\text{enc},i}^T, f_j)$ and $\cos(W_{\text{dec},i}, f_j)$, respectively. We re-arrange the indices of the SAE latents to best align visually with true features.

## 3.1 FULLY INDEPENDENT FEATURES

We first study the case of a toy model with $N = 4$ independent features. Features 1-3 fire with probability 0.25, and feature 4 fires with probability 0.2. We plot the encoder / decoder cosine similarity with true features in Figure 2. When features fire independently, the SAE learns correct features regardless of the width of the SAE.

c

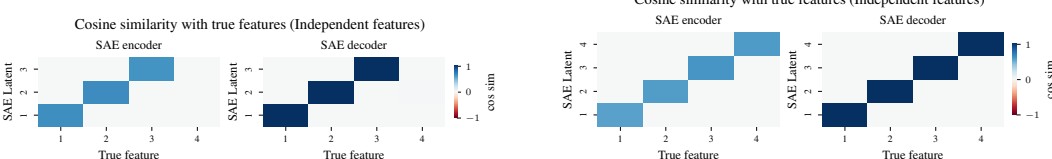

Figure 2: SAE with three latents (left) and four latents (right) trained on a toy model with independent features. Both SAEs learn correct features.

Unfortunately, real LLMs do not have fully independent features. SAEs were first studied under toy models with independent features (Elhage et al., 2022), and this is likely why the field was not aware of feature hedging much earlier.

## 3.2 HIERARCHICAL FEATURES

Next, we explore true features that fire hierarchically. We modify the toy model from Section 3.1 above, and set $f_3$ as the parent feature and $f_4$ as the child feature, so $f_4 \implies f_3$. That is, $f_4$ cannot fire unless $f_3$ is also firing. Hierarchical features cause feature absorption in SAEs that are wide enough to represent both the parent and child feature, but what happens if the SAE is not wide enough to represent the child latent? This is important as this is the intuition behind why Matryoshka SAEs work to combat absorption: if inner SAE levels are too narrow to represent both parent and child features, we hope that only the parent will be represented. We show results in Figure 3.

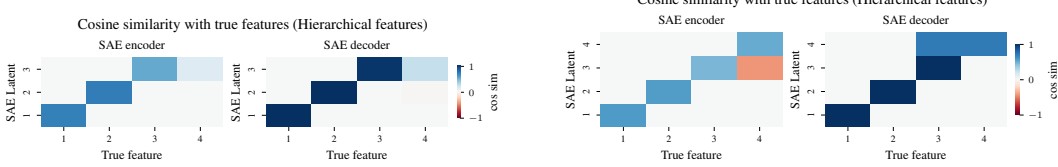

Figure 3: SAEs trained on a toy model with hierarchical features ($f_3 \implies f_4$). When the SAE is too narrow to represent $f_4$ (left), we see hedging where latent 3 mixes . When the SAE is wide enough to contain both $f_3$ and $f_4$ (right), we see feature absorption.

As expected, in the full-width SAE we see a classic feature absorption pattern. The parent latent encoder, $l_3$, learns $\neg f_4 \wedge f_3$, disabling the latent from firing if $f_4$ is present. The child latent, $l_4$, mixes both $f_3$ and $f_4$ together in the decoder.

However, when the SAE is too narrow to represent $f_4$, we now see that latent $l_3$ mixes a component of $f_4$ into both its encoder and decoder! We refer to this as *feature hedging*. The SAE is learning an incorrect, polysemantic latent that mixes correlated features.

While we expect this will be a problem for all SAEs, it is particularly problematic for Matryoshka SAEs, as Matryoshka SAEs combat absorption by using inner SAE levels that are too narrow to contain both parent and child features. However, as we see here, this causes hedging.

We study hierarchical features further in a single-latent SAE in Appendix A.1.2. We further show that MSE directly causes hedging with hierarchical features in Appendix 3.5.

### 3.3 POSITIVELY CORRELATED FEATURES

Hierarchy is a particularly extreme form of positive correlation, where a feature can only fire if another feature also fires. Next, we relax that restriction, and investigate what happens if a feature is merely more likely to fire along with another feature, but can still fire on its own as well. We modify the toy model so that $p_4 = 0.2$ if $f_3$ fires, but $p_4 = 0.1$ if $f_3$ does not fire, adding a small positive correlation between $f_3$ and $f_4$. We show results Figure 4

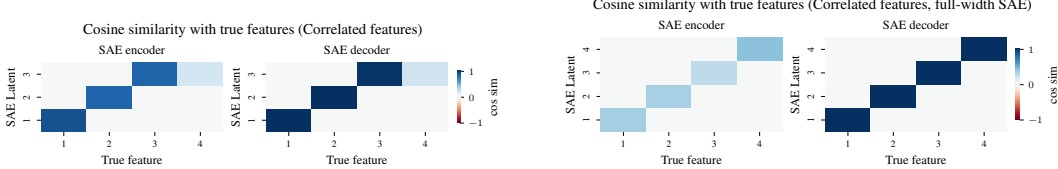

Figure 4: SAEs trained on a toy model with positive correlation between features $f_3$ and $f_4$. When the SAE is too narrow to represent $f_4$ (left), we still see hedging in latent 3. When the SAE is wide enough to contain both $f_3$ and $f_4$ (right), the SAE learns correct features.

We still see that the SAE is mixing a positive component of $f_4$ into $l_3$, despite there no longer being perfect hierarchy! We also see that if we extend the SAE width so that $f_4$ is tracked by its own latent $l_4$, there is now no absorption at all, as absorption requires (nearly) hierarchical features to arise.

### 3.4 ANTI-CORRELATED FEATURES

So far we have only seen the effect of positive correlation between features. We next change our toy model so $f_4$ is more likely to fire if $f_3$ does *not* fire. We set $p_4 = 0.1$ if $f_3$ fires, but $p_4 = 0.2$ if $f_3$ does not fire. Results are shown in Figure 5.

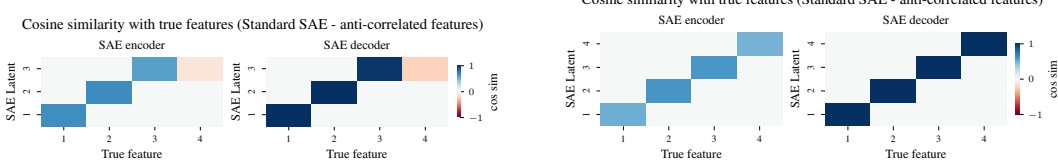

Figure 5: SAEs trained on a toy model with negative (anti) correlation between features $f_3$ and $f_4$. When the SAE is too narrow to represent $f_4$ (left), we latent 3 mixes a *negative* component of $f_4$. When the SAE is wide enough to contain both $f_3$ and $f_4$ (right), the SAE learns correct features.

We now see that $l_4$ is mixing in a *negative* component of $f_3$. This demonstrates that the correlation is the cause of the hedging: flipping the sign of the correlation flips the sign of the hedging. We study the mechanics of this phenomenon in more depth in Appendix A.1.

The implications of this for SAE performance are quite dire. While it is already bad for positively correlated features to become hedged (e.g. "sunshine" and "summertime"), at least the mixed features have some relation to each other. For anti-correlated features, this could look like a latent for "chemical molecules" having a negative component of the "Darth Vader" feature mixed in, since chemical modules and Darth Vader are highly anti-correlated. Worse, it is not even clear if the inverse of the "Darth Vader" direction is meaningful in the model at all, or is just noise. We further expect that there are many more negative correlations than positive correlations in language, e.g. a negative component every word in every non-English language may be mixed into every latent tracking an English word. These negative correlations likely introduce what looks like a lot of random noise into SAE latents, and this can only harm performance and interpretability.

### 3.5 HEDGING IS CAUSED BY RECONSTRUCTION LOSS

What causes hedging? We hypothesize that it is a combination of not enough latents to represent every feature, and the fact that MSE loss incentivizes reconstructing multiple features imperfectly as opposed to only one feature perfectly.

To test this, we analyze the loss curves for a single-latent tied SAE with a parent-child relationship between the two features $f_1$ and $f_2$, so $f_2 \implies f_1$. The ideal SAE latent must be some combination of these two features. As there are no other interfering features to break the symmetry between encoder and decoder, the SAE can be expressed by a single unit norm latent. We set the SAE latent $l$ to an interpolation of these two features, $l = \alpha f_2 + (1 - \alpha)f_1$ (adjusted to have unit norm). We calculate expected SAE loss consisting of MSE + L1 loss for $0 \leq \alpha \leq 1$.

First, we set $P(a = f_1) = 0.3$ and $P(a = f_1 + f_2) = 0.1$. We characterize the probabilities this way since there are only two firing possibilities we need to consider: either $f_1$ is firing on its own or $f_1$ and $f_2$ are firing together. We use L1 coefficient of 0 and 0.1 to explore the effect of the sparsity penalty on loss. We also consider the case where both features fire together more than they fire on their own, with $P(a = f_1) = 0.1$ and $P(a = f_1 + f_2) = 0.3$. Loss curves are shown in Figure 6.

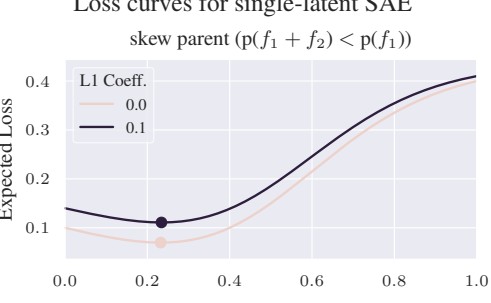
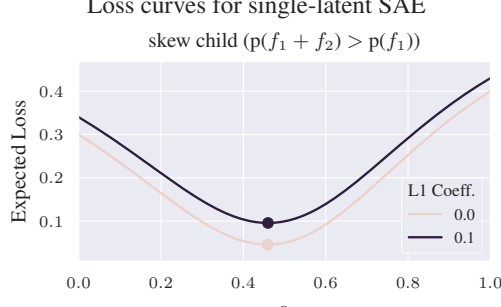

(a) Loss curves when the parent feature $f_1$ fires more on its own than with child feature $f_2$. Loss is minimized between $f_1$ and $f_2$ rather than at $f_1$ ($\alpha = 0$). Sparsity penalty does not change the minimum.

(b) Loss curves when the parent feature $f_1$ fires less on its own than it does with the child feature $f_2$. Loss is incorrectly minimized between $f_1$ and $f_2$. Sparsity penalty does not change the minimum.

Figure 6: Loss curves for an SAE with a single latent $l$ and 2 hierarchical features, where $f_2 \implies f_1$. The minimum loss is indicated with a dot on each plot. $\alpha = 0$ means that $l = f_1$, and $\alpha = 1$ means $l = f_2$. In all cases, loss is minimized when the latent $l$ is a combination of $f_1$ and $f_2$.

In these plots, $\alpha = 0$ corresponds to the SAE latent being exactly $f_1$, and $\alpha = 1$ corresponds to the latent being $f_2$, and $\alpha = 0.5$ corresponds to $f_1 + f_2$. We clearly see that the SAE loss has a single minimum between $f_1$ and $f_1 + f_2$, showing that the MSE minimum is attained with feature hedging.

**Theoretical proof**  We provide a proof that MSE loss causes hedging when there are correlated features and the SAE is narrower than the number of true features in Appendix A.4.

# 4  QUANTIFYING HEDGING IN LLM SAEs

While we have demonstrated hedging in a synthetic setting, it remains a question how much hedging occurs in LLM SAEs. Based on our understanding of hedging in toy models, we expect that when a new latent is added to an SAE, this should "pull out" the component of the new feature from existing SAE latents, where it was previously hedged. Thus if hedging occurs, *the change in existing latents after a new latent is added should project onto that new latent*. If hedging did not exist, then adding a new latent should not have any effect on existing latents.

**Parent latents are learned before child latents**  A key assumption in Matryoshka SAEs is that if latents exist in a hierarchy, and the SAE is too narrow to represent both the parent and child, the SAE will learn the parent first. We feel this assumption is reasonable since parent latents, by definition, fire more frequently than child latents, so the SAE is incentivized to learn them first. See Appendix A.13 for more formal arguments.

This insight allows us to differentiate hedging from absorption. Under absorption, if a newly added latent is a child feature of an existing latent, then the encoder for the parent latent adds a negative component of the child latent to avoid firing when the child is active, but *the parent decoder latent remains unchanged*. This corresponds to adding $l_2$ to Figure 1a and arriving at Figure 1b. The decoder of $l_1$ (the parent) remains identical to before $l_2$ is added, except the *hedging from $f_2$ is removed*. Thus, changes to existing decoder latents cannot be absorption and must be due to hedging.

**Hedging degree**  Taking this into account, we define a metric called hedging degree, $h$. We take an existing SAE $s_0$ with $L$ latents and add $N$ new latents to the SAE. After adding these latents, we continue training the SAE and arrive at a new SAE, $s_1$, with $L + N$ latents. We also continue training $s_0$ on the same tokens that we train $s_1$ on to ensure that any difference between $s_0$ and $s_1$ is due only to the newly added latents. $W_{\text{dec}}^0$ refers to the new decoder of $s_0$, and $W_{\text{dec}}^1$ refers to the decoder of $s_1$. $W_{dec}$ is normalized so each latent has unit norm. We define the difference in the original $L$ latents between $s_0$ and $s_1$ as:

$$\delta_L = W_{\text{dec}}^1[0 : L] - W_{\text{dec}}^0[0 : L] \tag{5}$$

where $W_{\text{dec}}^1[L : L + N]$ refers to the newly added decoder latents. $W_{\text{rand}}[0 : N]$ refers to a decoder consisting of $N$ randomly initialized unit-norm latents. All decoders are normalized to have latents of unit norm. We define the projection of a vector $v$ onto a subspace spanned by $W$ as:

$$\text{Proj}(v, W) = W(W^T W)^{-1} W^T v \tag{6}$$

We expect that even if there were no hedging at all, simply due to noise, existing SAE decoder latents may undergo a change that has some small projection onto new added latents. We want to make sure that anything we quantify as hedging must be larger than what we would expect from random noise. Taking this into account, the hedging degree $h$ is then defined as:

$$h = \frac{1}{L} \sum_i^L \underbrace{\|\text{Proj}(\delta_L[i], W_{\text{dec}}^1[L : L + N])\|_2}_{\text{Projection of } \delta_L \text{ onto N new latents}} - \underbrace{\|\text{Proj}(\delta_L[i], W_{\text{rand}}[0 : N])\|_2}_{\text{Projection of } \delta_L \text{ onto N random latents}} \tag{7}$$

Any value of $h > 0$ corresponds to hedging above what we would expect from random noise, as $h$ subtracts the projection along $N$ randomly initialized unit-norm latents as part of the computation.

The choice of the number of new latents $N$ is a hyperparameter of hedging degree. We use $N = 64$ for our hedging degree calculation. We explore the effect of different choices on $N$ in Appendix A.7. We validate this metric on toy models in Appendix A.5.

## 4.1  RESULTS

We experiment with SAEs trained on Gemma-2-2b (Team et al., 2024), as this model is commonly used for SAE research due to the thoroughness of the Gemma Scope suite of SAEs (Lieberum et al.,

2024), as well as Llama-3.2-1b (Dubey et al., 2024) to validate results on another LLM. All SAEs are trained first on 250M tokens of the Pile uncopyrighted (Gao et al., 2020). After adding $N = 64$ latents, we continue training for another 250M tokens. The version of the SAE without latents added is also trained for another 250M tokens, so each SAE is trained for 500M tokens total. The pair of extended and non-extended SAEs is used to calculate hedging degree. SAE training details are in Appendix A.6. In all plots, "btk" means BatchTopK, and "l1" refers to standard ReLU SAEs with an L1 sparsity penalty.

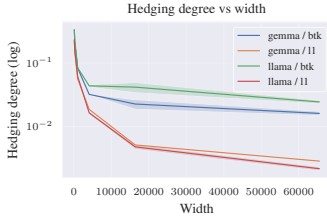
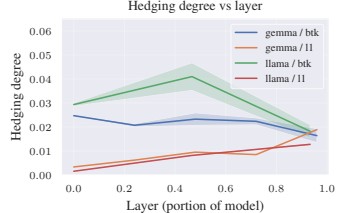
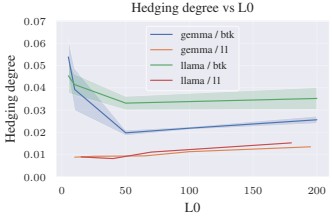

(a) Hedging degree vs width. No SAE tested reached 0 hedging.

(b) Hedging degree vs layer, normalized by number of LLM layers.

(c) Hedging degree vs L0.

Figure 7: Hedging degree for SAEs trained on Gemma-2-2b and Llama-3.2-1b. SAEs have width 8192 (except plot a), BatchTopK SAEs have K=25 (except plot c). Shaded area in plots is 1 std.

We first calculate hedging degree vs SAE width in Figure 7a, with widths ranging from 128 to 65536. Hedging degree is dramatically higher at narrower widths, especially at 4096 width and below. While the hedging rate drops a lot with increasing SAE width, even at our max width of 65536 no SAE achieves 0 hedging degree, indicating there is still hedging occurring.

We next calculate hedging degree vs L0 (the average number of active latents) in Figure 7c, with L0 ranging from about 5 to 200. Very low L0 seems to lead to more hedging for BatchTopK SAEs, but the effect is minor compared with the effect of SAE width on hedging degree.

Finally, we calculate hedging degree vs layer in Figure 7b. The hedging degree for L1 and TopK SAEs appears to merge around the end of the SAE, but overall the layer does not appear to have a massive effect on hedging degree.

It also appears that BatchTopK SAEs have more hedging than L1 SAEs. We suspect that L1 loss can reduce hedging from positively correlated features. We explore this further in Appendix A.1.3.

We further validate hedging in LLM SAEs via a case-study of adding a new latent to an SAE trained on Gemma-2-2b in Appendix A.8.

## 5 BALANCING HEDGING AND ABSORPTION IN MATRYOSHKA SAEs

Matryoshka SAEs (Bussmann et al., 2025) combat absorption with nested SAE loss prefixes. Each level acts like a small SAE, and is forced to reconstruct the input on its own. This forces the SAE to learn more general concepts in earlier levels, and makes it difficult for the SAE to make holes in the recall of parent latents for absorption, as this would hurt the reconstruction of earlier levels.

However, since early matryoshka levels are effectively narrow SAEs, they suffer from feature hedging. As we saw in Section 4.1, the more narrow an SAE is, the worse the hedging. Matryoshka SAEs thus solve feature absorption at the expense of exacerbating feature hedging.

Inspecting the effect of hedging and absorption on the SAE encoder in Figure 1b shows that hedging and absorption have opposite effects. For hierarchical features, hedging adds a positive component of child features into the parent encoder latent, but absorption does the opposite and adds a negative component of child features into the parent latent. If we balance the negative component of child latents from absorption with the positive component from hedging, these effects can cancel out.

**Balance matryoshka SAE** We extend the definition of a matryoshka SAE from Equation 4 to allow applying a scaling coefficient $\beta_m$ to the loss for each matryoshka level:

$$\mathcal{L} = \sum_{m \in \mathcal{M}} \beta_m \left( \|a - \hat{a}_m\|_2^2 + \lambda \mathcal{S}_m \right) + \alpha \mathcal{L}_{\text{aux}} \tag{8}$$

We refer to this extension as a *balance matryoshka SAE*, where each $\beta_m \geq 0$ controls the relative balance of each level. If each $\beta_m = 1$ this is a standard matryoshka SAE. If $\beta_m = 0$ for all matryoshka levels except the outer-most level, this reduces to a standard (non-matryoshka) SAE.

We demonstrate this balancing in a toy model of hierarchical features. The toy model has 4 features, with feature 1 being the parent feature and features 2-4 being children (features 2-4 can only fire if feature 1 is also firing). Feature 1 fires with probability 0.25, and each child feature fires with probability 0.15 if feature 1 is firing. We train a matryoshka SAE with a single inner level consisting of only latent 1 with balance coefficient $\beta$ (Since there is only one inner level, we always set the outer level coefficient to 1). For more details on this toy setup, see Appendix A.10.

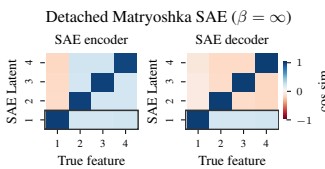 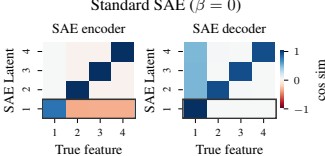 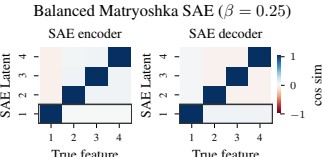

(a) Matryoshka SAE with detached loss (equivalent to a matryoshka SAE with $\beta = \infty$). Hedging adds positive components of the child features 2-4 to the encoder of latent 1.

(b) Standard SAE (equivalent a matryoshka SAE with $\beta = 0$). Absorption adds negative components of the child features 2-4 to the encoder of latent 1.

(c) Roughly balanced matryoshka SAE with $\beta = 0.25$. The positive and negative contributions hedging and absorption roughly cancel out, leaving a nearly perfect SAE.

Figure 8: Balancing hedging and absorption in a toy model of hierarchical features. Child features 2-4 only fire if parent feature 1 fires. The matryoshka SAE has a single inner level with 1 latent, represented by a black box around latent 1.

We show results in Figure 8. When $\beta$ is too high or too low this results in hedging or absorption, respectively. When $\beta = 0.25$, these balance out and the SAE learns a near perfect representation.

Next, we train LLM balance matryoshka SAEs with different balance ratios on Gemma-2-2b layer 12. The SAEs are BatchTopK with k=40, trained on 500M tokens. The SAEs have 5 matryoshka levels of sizes 128, 512, 2048, 8192, and 32768 (so the full SAE has width 32768). We set the outermost $\beta_5 = 1$, and set a constant multiplier between each subsequent $\beta_m$, so multiplier = $\beta_m/\beta_{m+1}$. If the multiplier is 0.5, then $\beta_m = 0.5^{(5-m)}$.

We train 10 seeds for each multiplier and show results in Figure 9 for absorption rate, targeted probe pertubation (TPP), Spurious Concept Removal (SCR), K-sparse probing, and feature-splitting metrics from SAEBench (Karvonen et al., 2025), and k=1 sparse probing results (Gurnee et al., 2023) for a Parts of Speech (POS) dataset we created using Treebank POS tagged sentences (Marcus et al., 1993). We add a POS dataset for probing since POS are very general concepts, and should be learned in the earliest levels of a matryoshka SAE.

For TPP, feature splitting, and sparse probing, using a compound multiplier of around 0.75 achieves better results than either a standard matryoshka SAE or a standard (non-matryoshka) SAE, providing evidence that balancing matryoshka losses can improve the performance. Using a multiplier of 0.75 still scores well on the absorption metric as well. Strangely, SCR appears to perform better at higher multipliers. However, SCR is also the noisiest metric, and the noise is higher at high multipliers, so it could be that hedging increases the noise of the SCR metric but does not fully break it. We provide further results and more details in Appendix A.12.

While balancing each $\beta_m$ can improve performance on most metrics, we do not expect this to perfectly solve absorption and hedging. We show in Appendix A.11 that balancing all hedging and absorption with a single $\beta_m$ is not always possible. We expect it may be possible to further improve performance by learning different balancing coefficients per latent, but this is left to future work.

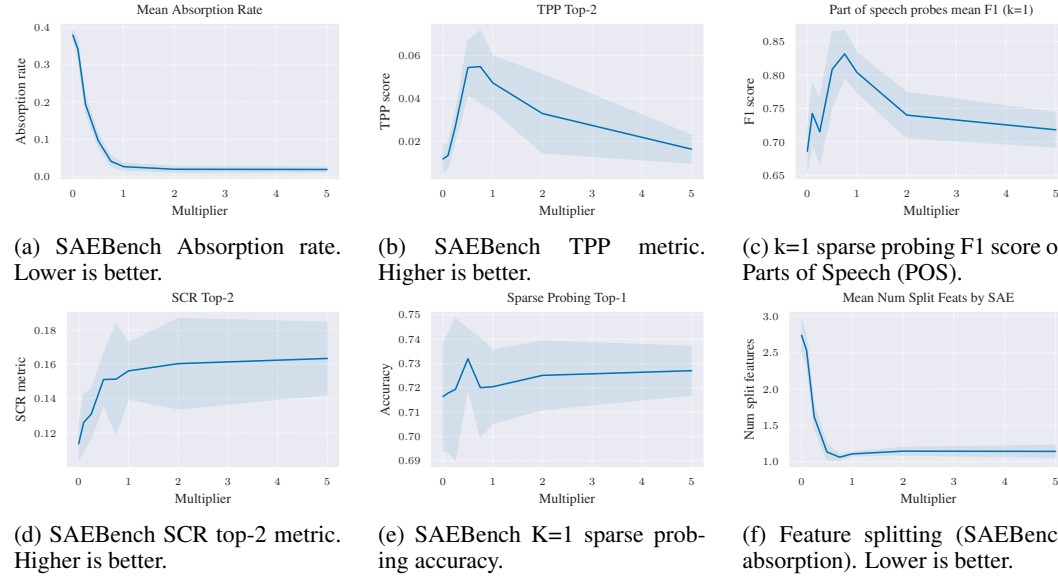

(a) SAEBench Absorption rate. Lower is better.

(b) SAEBench TPP metric. Higher is better.

(c) k=1 sparse probing F1 score on Parts of Speech (POS).

(d) SAEBench SCR top-2 metric. Higher is better.

(e) SAEBench K=1 sparse probing accuracy.

(f) Feature splitting (SAEBench absorption). Lower is better.

Figure 9: Performance of balance matryoshka SAEs vs multiplier. The shaded area is 1 std. Multiplier=0 is equivalent to a standard SAE, and multiplier=1 is a standard matryoshka SAE.

## 6 RELATED WORK

Other work has highlighted theoretical problems with SAEs. Till (2024) investigated a problem where SAEs may increase sparsity by inventing features. For instance, an SAE may fabricate a "red triangle" feature in addition to "red" and "triangle" features. Templeton et al. (2024) dicuss the problem of feature splitting, where an SAE may not learn features at a desired level of specificity. Engels et al. (2024) investigates SAE errors and finds that SAE error may be pathological and non-linear. Engels et al. (2025) further shows that there are features that cannot be expressed as a simple linear direction, and thus SAEs may struggle to represent these features. Wu et al. (2025) and Kantamneni et al. (2025) both investigate the empirical performance of SAEs and find that SAEs underperform baselines. In the field of complete dictionary learning, it has been shown that if features are highly coherent (correlated), dictionary learning will struggle to correctly identify them (Hu & Huang, 2023; Wang et al., 2020; Gribonval & Schnass, 2010). However, these works do not address the narrow regime ($L_{\text{SAE}} < L_{\text{true}}$) nor the failure modes from correlated features in this regime which we identify as feature hedging.

## 7 DISCUSSION

SAEs remain a promising technique for decomposing the residual stream of LLMs in an unsupervised manner. However, given recent work showing that SAEs underperform relative to baselines (Wu et al., 2025; Kantamneni et al., 2025), it is imperative that we understand the reasons for this underperformance so they can be addressed.

In this work, we introduced the problem of feature hedging in SAEs, showing it both theoretically in toy models, and empirically in SAEs trained on real LLMs. We suspect that hedging, along with absorption, may be one of the core theoretical problems leading to poor SAE performance.

Using our understanding of hedging, we introduced the balance matryoshka SAE architecture, allowing balancing of hedging and absorption against each other, improving interpretability. We view balance matryoshka SAEs as a starting point, and expect this architecture can be improved by optimizing the balance coefficients. There may not be a single coefficient that perfectly balances hedging and absorption for all features, so we expect there may be further gains from learning a different balancing coefficients per latent in the SAE. We leave these improvements to future work.

## 8 REPRODUCIBILITY STATEMENT

Code for all toy model experiments and demonstration code for training and evaluating LLM SAEs is provided as part of the supplementary materials for this paper. We further provide details on toy model SAE training in Section 3 and Appendix A.1. LLM SAE training is further detailed in Section 4.1 and Appendix A.6.

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

## A APPENDIX

### A.1 STUDYING HEDGING IN SINGLE-LATENT SAES

We begin by investigating hedging in the simplest possible toy SAE setting: an SAE with a single latent. We use a model with two true features $f_1$ and $f_2$ ($N = 2, D = 50$). Each feature fires with magnitude 1.0. Unless otherwise specified, $f_1$ fires with probability 0.25, and $f_2$ fires with probability 0.2. We use SAELens (Bloom et al., 2024) to train a single-latent SAE on these activations.

#### A.1.1 FULLY INDEPENDENT FEATURES

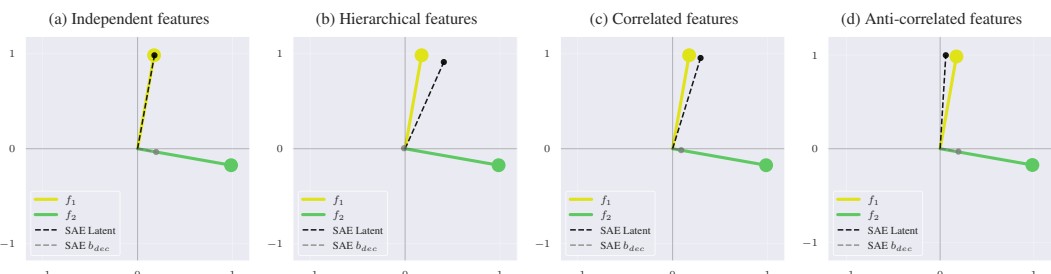

Figure 10: True features, and SAE decoder latent and $b_{\text{dec}}$ for single-latent SAE and a toy model with two true features. When the features fire independently, there is no hedging seen in the SAE latent. When any correlation is present, the SAE latent shows clear hedging.

We first study the case when $f_1$ and $f_2$ fire independently. We find that the SAE correctly represents $f_1$ without any interference from $f_2$. However, the decoder bias has incorrectly learned to represent the direction of $f_2$, but with magnitude 0.2, equal to the probability of $f_2$ firing. The single SAE latent, SAE bias term, and true features are shown in Figure 10a.

We consistently find this pattern of the decoder bias merging in positive components of features not tracked by their own latent. In this sense, the decoder bias can be thought of as an always-on latent, and thus is thus also susceptible to hedging.

#### A.1.2 HIERARCHICAL FEATURES

Next, we investigate what happens if $f_1$ and $f_2$ are in a hierarchy, so $f_2$ can only fire if $f_1$ fires, but $f_1$ can still fire on its own ($f_2 \implies f_1$). We adjust the firing probability of $f_2$ so that $P(f_2|f_1) = 0.2$, and $P(f_2|\neg f_1) = 0$ (thus, $P(f_2) = 0.05$). In a two-latent SAE this setup would cause feature absorption. We plot the SAE latent, decoder bias, and true features in Figure 10b.

Here we clearly see feature hedging. The single SAE latent has now merged in a component of $f_2$ into its single latent, so it is now a mixture of $f_1$ and $f_2$. This merging of features reduces the MSE loss of the SAE despite being a degenerate solution.

Increasing the L1 penalty of the SAE cannot solve this problem. $f_2$ only fires if $f_1$ fires, so adding a positive component of $f_2$ into the encoder does not cause the latent to fire any more often.

#### A.1.3 POSITIVELY CORRELATED FEATURES

Next, we change our setup so that $P(f_2|\neg f_1) = 0.1$ instead of 0. We still keep $P(f_2|f_1) = 0.2$, so that $f_2$ is more likely to fire if $f_1$ fires, but it can still fire on its own as well. The features are now merely correlated rather than following a strict hierarchy. Results are shown in Figure 10c.

We still see hedging in the SAE latent, but less than with full hierarchical features. However, if the L1 penalty is high enough and the level of correlation is low enough, then the SAE can still learn the correct features, as positive hedging increases the L0 of the

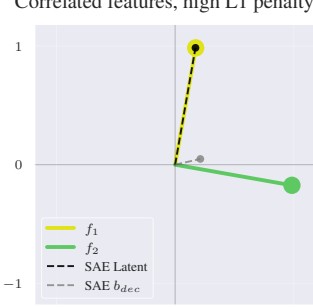

Figure 11: High L1 penalty can reduce hedging caused by positive correlations.

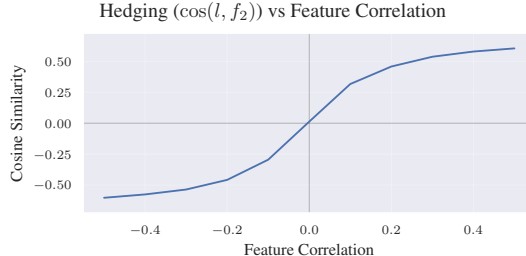

Figure 12: Hedging amount ($\cos(l, f_2) = 0$) vs correlation between $f_1$ and $f_2$. The amount of hedging is a clear function of the amount of correlation between feautres.

SAE slightly relative to learning just $f_1$. We show the resulting SAE latent and features with high L1 penalty in Figure 11. Interestingly, we now see that the hedging has moved more apparently into the decoder bias instead. If we use a full-width SAE, the SAE learns the true features despite the correlation (Appendix A.3).

### A.1.4   ANTI-CORRELATED FEATURES

Next, we reverse the conditional probabilities of $f_2$ so that $P(f_2|f_1) = 0.1$ and $P(f_2|\neg f_1) = 0.2$. Now $f_2$ is more likely to fire on its own than it is to fire along with $f_1$. Results are shown in Figure 10d.

Now the SAE latent has actually merged a *negative* component of $f_2$ into its single latent instead of a positive component. How does this work? We see that the decoder bias, $b_{\text{dec}}$, has a larger component of $f_2$ than in the positive correlation case. The SAE is using the decoder bias to include a "default" value for $f_2$, and then when $f_1$ fires, the SAE latent's negative component of $f_2$ acts to reduces the amount of $f_2$ present in the reconstruction. The SAE is abusing the correlation to adjust its guess of the amount of $f_2$ that should be output despite not having a dedicted latent for $f_2$: if $f_1$ is active, then the likelihood that $f_2$ is active decreases, and the SAE likewise reduces the amount of $f_2$ that is output.

Increasing L1 penalty cannot solve this, as the negative component of hedging in the encoder does not increase L0 of the SAE. If we use a full-width SAE, we again see the SAE learns the true features despite the correlation (see Appendix A.3).

### A.2   HEDGING IS A FUNCTION OF FEATURE CORRELATION

Next, we explore the effect of feature correlation on the amount of hedging in our single-latent, two feature setting. We set $P(f_1) = 0.45$ and $P(f_2) = 0.25$, but change the correlation between these features, $\rho$, to range from $-0.5$ to $0.5$. We then calculate the cosine similarity of the SAE decoder latent, $l$, with $f_2$. We furthermore initialize the single SAE latent to match $f_1$, so that any deviation from this must be caused by gradient pressure rather than simply being an unfortunate local minimum. If there is no hedging occurring, then $\cos(l, f_2) = 0$, as we saw in Figure 10a. Results are shown in Figure 12.

As expected, the amount of hedging directly tracks the amount of correlation. The hedging also matches the sign of the correlation as well, with negative correlation resulting in a negative component of $f_2$ being mixed into $l$, and positive correlation resulting in a positive component of $f_2$ being mixed into $l$.

### A.3   FULL-WIDTH SAE TOY MODEL RESULTS

We extend the discussion of single-latent SAEs to explore what happens if the SAE has two latents, the same number of latents as the number of true features. We use the same toy model as in Section A.1.3 for the positive correlation case, and the same toy model as in Section A.1.4 for the

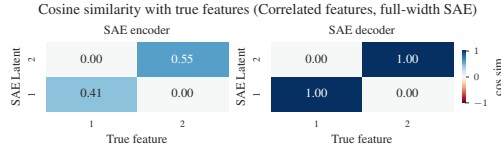 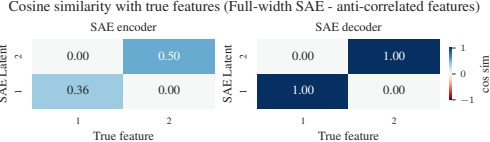

(a) Full-width SAE with correlated features. The SAE is still able to perfectly learn the underlying features despite the correlation.

(b) Full-width SAE with anti-correlated features. The SAE is still able to perfectly learn the underlying features despite the correlation.

Figure 13: Full-width SAE results on correlated and anti-correlated toy models.

anti-correlated case. We use L1 penalty of 1e-3 for the positive correlation case, the same as the L1 penalty that caused hedging in single-latent SAEs.

We plot the results in Figure 13. In both cases, the full-width SAEs are able to perfectly recover the true features despite the correlation, and despite the low L1 penalty. This shows that hedging is caused by the SAE being too narrow, as increasing the width of the SAE solves the problem.

## A.4 THEORETICAL DERIVATION: FEATURE HEDGING IN MSE-OPTIMAL SPARSE AUTOENCODERS

**Theorem.** *Consider a generative model with $N$ orthogonal features where feature $j$ fires with probability $p_j$ and magnitude $m_j \geq 0$. Let a Sparse Autoencoder (SAE) with capacity $M \leq N$ be trained to minimize Mean Squared Error (MSE). To isolate the behavior of the decoder, we assume the SAE possesses a **perfect oracle encoder** for the first $M$ features (recovering both activation and magnitude). The optimal reconstruction weight $V_{ik}$ for an untracked "orphan" feature $f_k$ ($k > M$) in latent $z_i$ is determined by the partial correlation between $f_i$ and $f_k$, scaled by their relative magnitudes. Specifically:*

1. **Hedging:** If $f_k$ is correlated with $f_i$ (e.g., hierarchical), $V_{ik} \neq 0$.

2. **Monosemanticity:** If $f_k$ is independent of $f_i$, $V_{ik} = 0$.

3. **Bias:** The decoder bias captures the mean of $f_k$ not explained by the hedged latents.

### A.4.1 PROBLEM SETUP

Let the input data $x \in \mathbb{R}^d$ be generated by a set of $N$ mutually orthogonal unit vectors $\{f_1, \ldots, f_N\}$. We define the data generation process as:

$$x = \sum_{j=1}^{N} A_j m_j f_j \tag{9}$$

Where:

- $A_j \sim \text{Bernoulli}(p_j)$ is the binary activation of feature $j$.

- $m_j \geq 0$ is the scalar magnitude of feature $j$.

- The features are orthogonal: $f_i^\top f_j = \delta_{ij}$.

A standard SAE consists of an encoder $z = \sigma(W_{enc}(x - b_{dec}) + b_{enc})$ and a linear decoder. For this derivation, we abstract away the specific implementation of the encoder (e.g., ReLU vs. TopK) to prove that hedging is optimal even in the best-case scenario.

We define the reconstruction $\hat{x}$ using an augmented latent vector $\mathbf{z} = [z_0, z_1, \ldots, z_M]^\top \in \mathbb{R}^{M+1}$, where $z_0 \equiv 1$ represents the learned decoder bias. We seek a decoding matrix $V \in \mathbb{R}^{(M+1) \times N}$ such that:

$$\hat{x} = \sum_{i=0}^{M} z_i \left( \sum_{j=1}^{N} V_{ij} f_j \right) \tag{10}$$

Here, $V_{ij}$ represents the weight of feature $f_j$ in the direction of latent $l_i$.

### A.4.2 ASSUMPTIONS

To analytically solve for the optimal weights, we make the following simplifying assumptions.

1. **Perfect Oracle Encoder:** We assume the encoder is an oracle that perfectly recovers the **scalar activation** (magnitude included) of the features it is assigned to track. Instead of deriving $z$ from $x$, we fix:
$$z_i = A_i m_i \quad \text{for } 1 \le i \le M$$
This ensures that any hedging observed in the decoder is driven by the loss landscape, not by encoder errors.

2. **Decoder Bias as Latent:** We treat the decoder bias as an "always-on" latent $z_0 \equiv 1$.

### A.4.3 THE MINIMIZATION PROBLEM

We seek to minimize the expected Mean Squared Error (MSE):
$$\mathcal{L} = \mathbb{E}[\|x - \hat{x}\|^2] \tag{11}$$

Since the features $f_j$ are orthogonal, the MSE loss decomposes into a sum of independent squared errors for the reconstruction of each feature. We can therefore solve for the optimal weights for any specific feature $f_k$ independently.

For a specific feature $f_k$, the loss is:

$$\mathcal{L}_k = \mathbb{E}\left[ \left( A_k m_k - \sum_{i=0}^{M} z_i V_{ik} \right)^2 \right] \tag{12}$$

Given Assumption 1 ($z_i = A_i m_i$), this minimization is equivalent to a **Linear Regression** (Ordinary Least Squares) of the target variable $Y = A_k m_k$ onto the regressors $\mathbf{z} = [1, A_1 m_1, \dots, A_M m_M]^\top$.

### A.4.4 DERIVATION OF OPTIMAL WEIGHTS

**Reconstructing Tracked Features ($k \le M$)** For any feature $f_k$ that has a dedicated latent $z_k = A_k m_k$, the regression is trivial. The latent $z_k$ is a perfect predictor of the target $A_k m_k$ with a coefficient of 1.
$$V_{kk} = 1, \quad V_{ik} = 0 \text{ for } i \ne k \tag{13}$$
**Result:** Tracked features remain monosemantic with unit weight (the magnitude is carried by the latent $z_k$).

**Reconstructing Untracked "Orphan" Features ($k > M$)** For an "orphan" feature $f_k$ that the SAE does not track explicitly, it must use the existing latents and bias to approximate it. The optimal weights are determined by the multivariate regression coefficients.

Let $\mathbf{z}_{1:M} = [z_1, \dots, z_M]^\top$ be the vector of tracked latents. Let $\Sigma_{\mathbf{z}} = \text{Cov}(\mathbf{z}_{1:M})$ be the covariance matrix of the latents. Let $\mathbf{c}_k = \text{Cov}(\mathbf{z}_{1:M}, A_k m_k)$ be the vector of covariances between the tracked latents and the orphan target.

The optimal weight vector $\mathbf{v}_k = [V_{1k}, \dots, V_{Mk}]^\top$ for the latents and the bias $V_{0k}$ are given by the standard OLS solutions:

$$\mathbf{v}_k = \Sigma_{\mathbf{z}}^{-1} \mathbf{c}_k \tag{14}$$

$$V_{0k} = \mathbb{E}[A_k m_k] - \mathbf{v}_k^\top \mathbb{E}[\mathbf{z}_{1:M}] \tag{15}$$

A.4.5 PROOF OF HEDGING CONDITIONS

To build intuition, we consider the case where the tracked latents are mutually independent (making $\Sigma_{\mathbf{z}}$ diagonal). In the general case where tracked latents are correlated, the optimal weights would be determined by the partial correlations (via the inverse covariance matrix $\Sigma_{\mathbf{z}}^{-1}$). Under the independence assumption, the matrix solution decouples into scalar solutions for each weight $V_{ik}$:

$$V_{ik} = \frac{\text{Cov}(z_i, A_k m_k)}{\text{Var}(z_i)}$$

Since $z_i = A_i m_i$, we can factor out the magnitudes:

$$V_{ik} = \frac{m_i m_k \text{Cov}(A_i, A_k)}{m_i^2 \text{Var}(A_i)} = \frac{m_k}{m_i} \frac{\text{Cov}(A_i, A_k)}{\text{Var}(A_i)}$$

This formula shows clearly that hedging is driven by correlation scaled by the **magnitude ratio** of the features.

**Case A: Independent Features (No Hedging)**  Assume the orphan $f_k$ is statistically independent of latent $f_i$. Then $\text{Cov}(A_i, A_k) = 0$.

$$V_{ik} = 0 \tag{16}$$

**Result:** The latent $l_i$ remains monosemantic ($V_{ik} = 0$). The reconstruction of the orphan feature is handled entirely by the decoder bias ($V_{0k} = p_k m_k$).

**Case B: Hierarchical Features (Hedging)**  Assume a strict hierarchy where the orphan is a child of the tracked feature: $f_k \implies f_i$. Thus, $A_k = 1 \implies A_i = 1$, so the covariance is $\text{Cov}(A_i, A_k) = p_k(1 - p_i)$. Substituting this into the equation:

$$V_{ik} = \frac{m_k}{m_i} \frac{p_k(1 - p_i)}{p_i(1 - p_i)} = \frac{m_k}{m_i} \frac{p_k}{p_i}$$

Recognizing that for hierarchical features $P(f_k|f_i) = p_k/p_i$, we obtain:

$$V_{ik} = \frac{m_k}{m_i} \cdot P(f_k|f_i) \tag{17}$$

**Result:** The latent $l_i$ becomes polysemantic. It learns a component of $f_k$ proportional to the conditional probability of the orphan given the parent, **scaled by the ratio of their magnitudes**.

**Case C: Anti-Correlated Features (Negative Hedging)**  Assume $f_k$ and $f_i$ are mutually exclusive ($f_k \cap f_i = \emptyset$). Then $\text{Cov}(A_i, A_k) = -p_i p_k$.

$$V_{ik} = \frac{m_k}{m_i} \frac{-p_i p_k}{p_i(1 - p_i)} = -\frac{m_k}{m_i} \frac{p_k}{1 - p_i} \tag{18}$$

We can further derive the exact value of the decoder bias $V_{0k}$ in this setting:

$$V_{0k} = \mathbb{E}[A_k m_k] - V_{ik}\mathbb{E}[z_i] = p_k m_k - \left(-\frac{m_k}{m_i} \frac{p_k}{1 - p_i}\right)(p_i m_i)$$

Simplifying the terms:

$$V_{0k} = p_k m_k \left(1 + \frac{p_i}{1 - p_i}\right) = m_k \frac{p_k}{1 - p_i} = m_k P(f_k|\neg f_i) \tag{19}$$

**Result:** The latent learns a **negative** component of the orphan feature. The decoder bias compensates by learning a large positive "default" value: precisely the probability of the orphan firing given the anti-correlated feature is *off*.

**Case D: Full Capacity** ($M = N$)   In the case where the SAE capacity equals the number of generative features ($M = N$), the set of orphan features $\{k \mid k > M\}$ is empty. Consequently, every feature $f_k$ falls under the case derived in the first paragraph of Section A.4.4.

$$V_{kk} = 1, \quad V_{ik} = 0 \text{ for } i \neq k \tag{20}$$

**Result:** When capacity is sufficient to track every feature ($M = N$), hedging vanishes entirely. The SAE learns a perfectly monosemantic representation.

### A.4.6   CONCLUSION

We have derived that under optimal MSE minimization with perfect scalar encoding:

1. **Feature Hedging** is the mathematically optimal strategy for reconstructing correlated features that exceed the model's capacity ($M < N$).

2. **Magnitude Ratios** play a crucial role: tracked latents hedge more aggressively towards high-magnitude orphan features.

3. **Independent features** do not cause hedging in the latents; their average presence is captured entirely by the **polysemantic decoder bias**.

4. **Correlated features** (hierarchical or anti-correlated) force the latents to rotate away from monosemanticity, mixing in components of untracked features based on their conditional probabilities.

### A.4.7   COROLLARY: SYMMETRIC ENCODER HEDGING

In the derivations above, we solved for the optimal decoder assuming a fixed encoder. However, in a real SAE trained end-to-end, both encoder and decoder are optimized simultaneously. We now show that if the decoder hedges, the encoder is mathematically compelled to hedge symmetrically.

**Theorem.** *Given a fixed decoder direction* $\mathbf{d}_i$ *that is "hedged" (a mixture of features), and assuming the encoder maintains* **perfect support selection** *(latent* $z_i$ *activates if and only if feature* $A_i$ *is active), the MSE-optimal linear encoder direction* $\mathbf{w}_i$ *must be collinear with* $\mathbf{d}_i$. *Thus, the encoder inherits the same polysemantic mixture as the decoder.*

**Proof**   Consider the reconstruction contribution of a single latent $z_i$ with a fixed decoder vector $\mathbf{d}_i$. Consistent with Assumption 1, we maintain the **perfect support recovery** assumption: the latent is active ($z_i \neq 0$) if and only if the target feature is active ($A_i = 1$). We seek the optimal encoder weight vector $\mathbf{w}_i$ that determines the activation value $z_i = \mathbf{w}_i^\top x$ when active.

The loss function, conditioned on the feature being active ($A_i = 1$), is:

$$\mathcal{L} = \mathbb{E}\left[\|x - (\mathbf{w}_i^\top x)\mathbf{d}_i\|^2 \mid A_i = 1\right] \tag{21}$$

This is a standard projection problem. We define the optimal scalar activation $z^*$ as the value that minimizes the distance between $x$ and the line spanned by $\mathbf{d}_i$. The solution is the orthogonal projection of $x$ onto $\mathbf{d}_i$:

$$z^* = \frac{x^\top \mathbf{d}_i}{\|\mathbf{d}_i\|^2} = x^\top \left(\frac{\mathbf{d}_i}{\|\mathbf{d}_i\|^2}\right) \tag{22}$$

Since our encoder is defined as $z_i = \mathbf{w}_i^\top x$, equating terms yields the optimal encoder weights:

$$\mathbf{w}_i^* = \frac{\mathbf{d}_i}{\|\mathbf{d}_i\|^2} \propto \mathbf{d}_i \tag{23}$$

**Conclusion:** The optimal encoder acts as a **matched filter** for the decoder.

1. From our previous derivation, we know the optimal decoder $\mathbf{d}_i$ is a mixture of the tracked feature $f_i$ and the orphan feature $f_k$.

2. Therefore, $\mathbf{w}_i^*$ must also be a mixture of $f_i$ and $f_k$.

This explains the empirical observation that hedging is **symmetric** (as seen in Figure 1a). The loss landscape forces the decoder to mix features to capture unrepresented variance, and simultaneously forces the encoder to mix features to detect that variance.

### A.5 VALIDATING HEDGING DEGREE IN TOY MODELS

To validate that our hedging degree metric works as expected, we set up a larger toy model with correlated features and train SAEs of varying widths on this toy model, calculating the hedging degree for each SAE. Our toy model consists of $N = 50$ mutually-orthogonal true features each with dimension $D = 100$. We randomly generate a correlation matrix to control feature firing correlations. Each feature $f_i$ fires with magnitude $m_i \sim \mathcal{N}(1.0, 0.15)$. We linearly decay the base firing probabilities $p_i$ from $p_0 = 0.345$ to $p_{49} = 0.05$ so that on average 11 features fire per input. The correlation matrix and feature firing probabilities are shown in Figure 14.

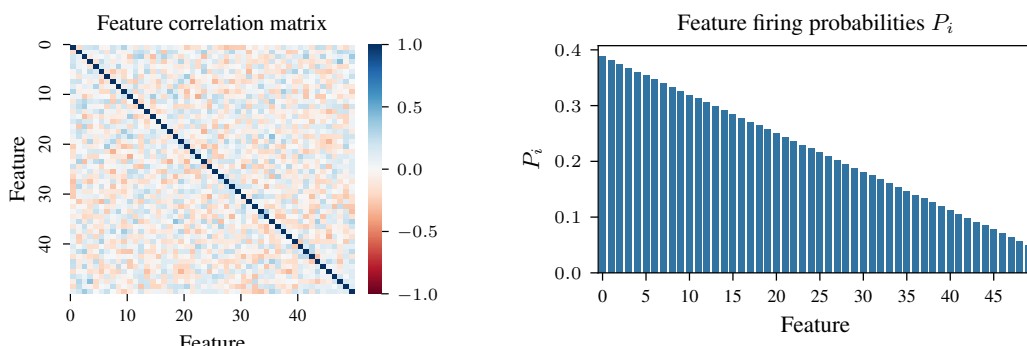

Figure 14: Feature correlation matrix (left) and firing probabilities (right) for our large toy model of correlated features.

We train a series of BatchTopK SAEs on activations generated by this toy setup. We follow the procedure in Section 4, first training a base SAE on 3M training samples. Then, we continue training a control variant of the SAE on another 3M samples, and an extended variants where we add 2 latents to the base SAE and train for 3M more samples. We then calculate the hedging degree for each of these SAEs. We set K=11 for the BatchTopK SAEs, matching the L0 of the underlying toy model. We plot results in Figure 15.

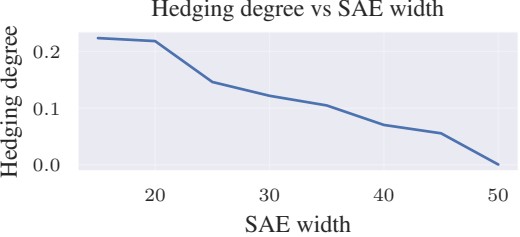

Figure 15: Hedging degree vs SAE width from our toy model. The narrower the SAE, the higher the hedging degree. When the SAE width equals the width of the toy model (50), hedging degree is zero.

We see that the narrower the SAE, the higher the hedging degree. Furthermore, once the SAE width matches the toy model width, hedging degree reaches zero.

### A.6 TRAINING DETAILS FOR LLM SAEs

All SAEs are trained on the Pile uncopyrighted (Gao et al., 2020), using a batch size of 4096 activations and context length of 1024 tokens. For L1 SAEs, we use a linear L1 warm-up of 10k steps. SAEs are trained on a single 80gb Nvidia H100 GPU. Model weights are loaded in fp32 precisions, but autocast to bfloat16 during training. We initialize the SAE so that the encoder and decoder are identical, where each latent has norm 0.1, following the procedure described in (Olah et al., 2024). All L1 SAEs are trained with learning rate 7e-5, and BatchTopK SAEs are trained with learning rate 3e-4. SAEs are trained using SAELens (Bloom et al., 2024).

Unless otherwise specified, BatchTopK SAEs use k=25. For SAEs trained on Gemma-2-2b, we conduct most experiments at layer 12 (roughly in the middle), and L1 SAEs trained on Gemma-2-2b use L1 coefficient of 10. This coefficient does not result in dead extension latents, and yields a L0 around 50. For SAEs trained on Llama-3.2-1b, we conduct most experiments at layer 7 (roughly in the middle of the model), and for L1 SAEs trained on Llama-3.2-1b, we use L1 coefficient of 0.5. This coefficient does not result in dead extension latents, and yields a L0 around 50.

### A.7 CHOICE OF HEDGING HYPERPARAMETER N

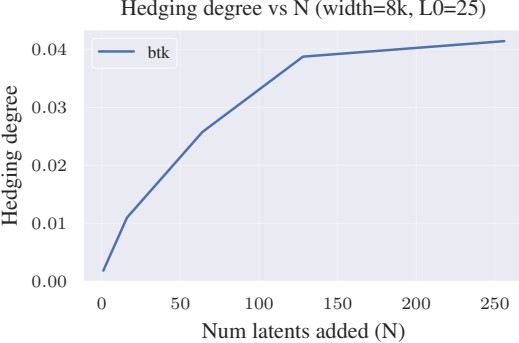

Figure 16: Hedging degree vs N

Our hedging degree metric requires adding N new latents onto an existing SAE to extend it, naturally leading to the question of what is a reasonable choice of N. We plot hedging degree vs N for Gemma-2-2b layer 12, given an initial BatchTopK SAE of width 8192 in Figure 16. We find that hedging degree increases until about N=250. We choose N=64 for our experiments, as 64 is still a small number of latents relative to the size of the residual stream (2304 for Gemma-2-2b), while still being large enough to hopefully reduce noise from any specific latent that gets added. Furthermore, as we see in the plot, the hedging degree from N=64 is about in the middle of the curve, further validating that this is a reasonable choice.

### A.7.1 EXTENDING LLM SAEs

We train two versions of extension SAEs - one for L1 loss SAEs and one for BatchTopK SAEs. In both cases, we begin with a pretrained SAE and add $N$ latents randomly initialized with norm 0.1, and with the same encoder and decoder directions, following Olah et al. (2024). For the BatchTopK SAEs, we simply train the SAE from this point as normal, as the TopK auxiliary loss (Gao et al., 2024) will naturally ensure that the newly added latents do not simply die off.

For L1 SAEs with high L1 penalty, dead latents become a more serious problem. We find that most of the newly added extension latents will rapidly be killed off if we simply train as normal. To combat this, we re-warm-up the L1 penalty. However, we cap the minimum L1 penalty at $\lambda_{\min}$, so for the portion of the warm-up where the L1 penalty would normally be below $\lambda_{\min}$, the L1 penalty is left at $\lambda_{\min}$ instead. This capping helps ensure the existing SAE latents are not very disturbed by this change in the L1 penalty. If the final L1 penalty is $\lambda_{\min}$ or below, then we do not perform this warm-up at all, as the L1 penalty is not strong enough to immediately kill off the newly added latents.

For Gemma-2-2b SAEs, we set $\lambda_{\min} = 10.0$. For Llama-3.2-1b SAEs, we set $\lambda_{\min} = 0.5$.

This warm-up procedure is only used for the high-L1 variants in Figure 7c - for all other plots the L1 coefficient used is less than $\lambda_{\min}$, so no warmup is needed.

## A.8 Case study: adding a new latent to an existing SAE

We next explore how hedging affects a real SAE. We trained a L1 SAE on Gemma-2-2b layer 12 with width 8192 for 250M tokens on the Pile (Gao et al., 2020), then add a new latent to the SAE, and continue training both the original SAE and the extended SAE for another 250M tokens.

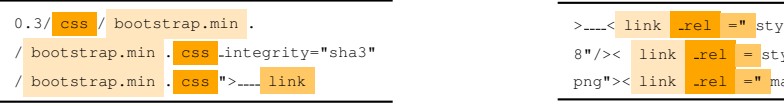

(a) Newly added case-study latent, latent 8192. The latent appears to track CSS scripts in HTML.

(b) Latent 3094, which had the largest negative $\delta$-projection after adding latent 8192. This latent tracks "rel" in HTML, used for CSS in HTML.

Figure 17: Sample top activating examples for case study latents.

We examine inputs that cause the newly added latent to fire to get a sense of what it represents. We reproduce a portion of the top activating examples for the new latent in Figure 17a. This latent appears to fire on CSS scripts included in HTML. A larger set of inputs is shown in Appendix A.9.

Next, we look at the magnitude of change in existing latents projected on the new latent. Based on our understanding of hedging, if a latent loses a large component of the newly added latent, this corresponds to a likely hierarchical relationship with the new latent. The latent which lost the largest component of the new latent is latent 3094, which seems to track the "rel" HTML attribute used mainly for linking CSS scripts. We show top activating examples for latent 3094 in Figure 17b.

Since CSS scripts are just one type of asset that can be linked using "rel", this appears to be exactly the sort of hierarchical relationship we expect to be heavily impacted by hedging.

## A.9 Additional case study dashboards

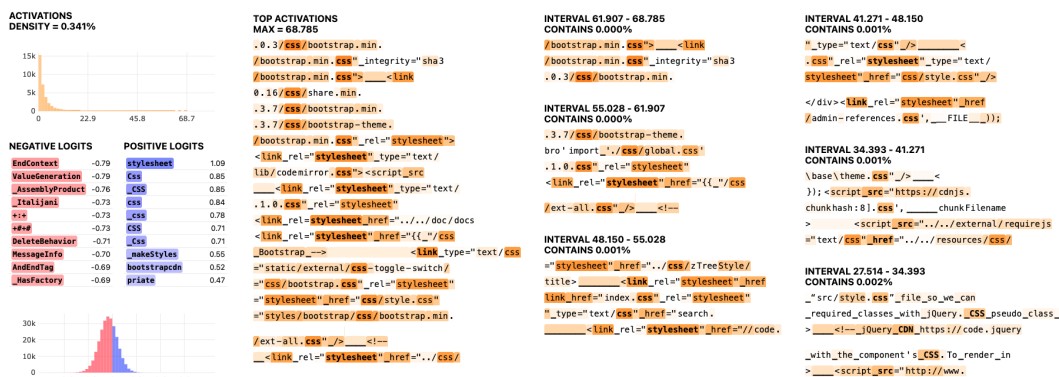

Figure 18: Dashboard for the newly added case study latent representing CSS scripts in HTML.

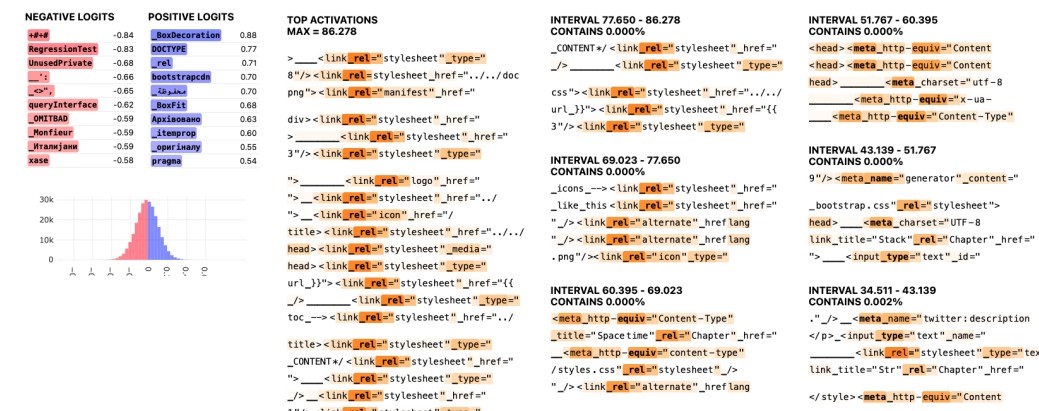

Figure 19: Dashboard for latent 3094, representing the "rel" HTML attribute used for CSS scripts. This latent has the highest negative δ-projection on the newly added case study latent.

## A.10 TOY BALANCE MATRYOSHKA SAEs

To explore the effect of balancing matryoshka losses in a simple toy setting, we create a toy model with 4 true features, all mutually orthogonal and with unit norm in a 50 dimensional space. We set up a hierarchical relationship between these features, so feature 1 fires with probability 0.25, and features 2, 3, and 4 all fire with probability 0.15 only if feature 1 fires. Thus, feature 1 is the parent feature in the hierarchy and features 2, 3, and 4 are all child features.

We train a matryoshka SAE with 4 latents on 100,000,000 samples from this toy model. The matryoshka SAE has a single inner level consisting of 1 latent, to match the number of parent latents in our hierarchy. Since our goal with this toy is just to build intuition, we initialize the SAE to the correct solution and allow the training to thus pull it away from this correct solution. This also ensures that each variation of our SAE with different balancing co-efficients learns the same latents in the same order, so visual comparison is easy.

## A.11 TOY UNBALANCEABLE MATRYOSHKA SAEs

The situation above where each child feature has the same probability of firing is unrealistic - we would expect that child features all fire with different probabilities from each other. Can we still balance the SAE perfectly in this situation? We adjust the toy model from above so that the 3 child features fire with probabilities 0.02, 0.2, and 0.5 for features $f_2$, $f_3$, and $f_4$, respectively. We then try to manually balance this SAE, finding that $\beta = 0.17$ gives roughly the best balance. We plots the resulting encoder/decoder cosine similarities in Figure 20.

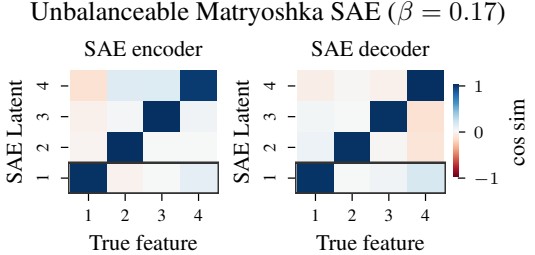

Figure 20: SAE encoder and decoder vs true feature cosine similarities for a balance matryoshka SAE where the child features fire with different probabilities. It's no longer possible to perfectly balance all 3 child features with the same $\beta$, but we can still do reasonably well.

We now see it is no longer possible to choose a single $\beta$ that perfectly balances all 3 children. We see slight hedging of feature 4 in latent 1, and slight absorption of feature 2 in latent 1. Still, this

looks decent compared to the full hedging or full absorption scenario, so we still expect that while balancing is not a perfect solution, it should be an improvement. We believe it should be possible to finding ways of better balancing the contribution of each outer latent on each inner latent, but this is left to future work.

### A.12 SAE EVALUATION

#### A.12.1 SAEBENCH EVALS

We evaluate our SAEs mainly using SAEBench (Karvonen et al., 2025). All evals are performed using default settings. We run all evaluations on an Nvidia H100 GPU with 80gb GPU memory. We evaluate on the following SAEBench tasks:

**K-sparse probing**  k-sparse probing builds on the work of Gurnee et al. (2023), where the goal is to create a linear probe from model activations using only $k$ neurons as input to the probe. This was adapted for use as an SAE evaluation technique by Gao et al. (2024). We focus mainly on $k = 1$ sparse probing, which means finding the single best SAE latent that serves as a classifier for a given concept, and evaluating the accuracy of that latent. SAEBench includes supervised classification datasets on which k-sparse probing is evaluated.

**Feature absorption**  The feature absorption metric in SAEBench is a variation on the metric defined in the original feature absorption work (Chanin et al., 2024). This metric uses a first-letter spelling task and first identifies the "main" latents for that task using k-sparse probing (Gurnee et al., 2023). Then, the metric identifies cases where a linear probe is able to correctly perform the first-letter classification task, but the "main" SAE latents fail to perform the task. The metric also looks for other latents that project onto the linear probe direction and fire in place of the "main' latents. Lower absorption is better.

The SAEBench absorption metric also includes "absorptions fraction", "feature splitting", and "first-letter k=1 sparse probing" results as well, which we include in our extended results. Absorption fraction detects partial absorption, where a parent latent can still fire but weaker when an absorbing child latent fires as well. Feature splitting detects the amount of interpretable feature splitting occurring in the SAE. Interpretable feature splitting is still considered negative, as we would prefer that features do not split at all and the SAE can still represent general, high-level concepts. The k-sparse probing results for the first-letter spelling task is calculated as part of the absorption metric, but is an interesting sparse-probing result in and of itself.

**Spurious concept removal (SCR)**  SCR builds on the SHIFT method from Marks et al. (2025) to detect how well an SAE isolates concepts. The metric uses datasets where two properties are perfectly entangled, for instance "profession" and "gender", and trains a biased probe on these concepts. The SCR metric then detects how well $k$ SAE latents can be ablated to de-bias the probe. If the SAE latents learn disentangled concepts, then it should only take a few SAE latents to perfectly de-bias the probe. A high SCR score means the SAE latents represent disentangled concepts.

**Targeted probe perturbation (TPP)**  The TPP metric extends SCR to multi-class labels. Binary probes are trained for each class, and TPP measures how well ablating $k$ SAE latents can degrade the performance of just one of the probes without degrading performance on the other probes. A high TPP score means that concepts are represented by distinct sets of SAE latents, rather than latents being entangled with many concepts.

#### A.12.2 PARTS OF SPEECH (POS) PROBING DATASET

We are interested as well in general, high-frequency concepts that we expect should be learned in the inner-most levels of a matryoshka SAE. These concepts should be the most affected by both absorption and hedging, as these concepts can be considered parent concepts to most other concepts. Parts of speech (POS) is a great test-case for these general concepts, and are not covered by the SAEBench sparse probing task. As such, we create our own custom POS dataset using the Penn Treebank tagged sentences (Marcus et al., 1993).

We simplify the Treebank parts of speech to the following core set:

"TO", "IN", "DT", "CC", "NNS", "PRP", "POS"

We pass these tagged sentences through an LLM, and then collect activations for the final token of position of each word at a given layer in the LLM. We create a binary classification dataset for each of these parts of speech, and perform k-sparse probing (Gurnee et al., 2023) on SAE latents to find the top k latents that act as a classifier for each of these parts of speech.

### A.12.3 BALANCE MATRYOSHKA SAE FULL RESULTS

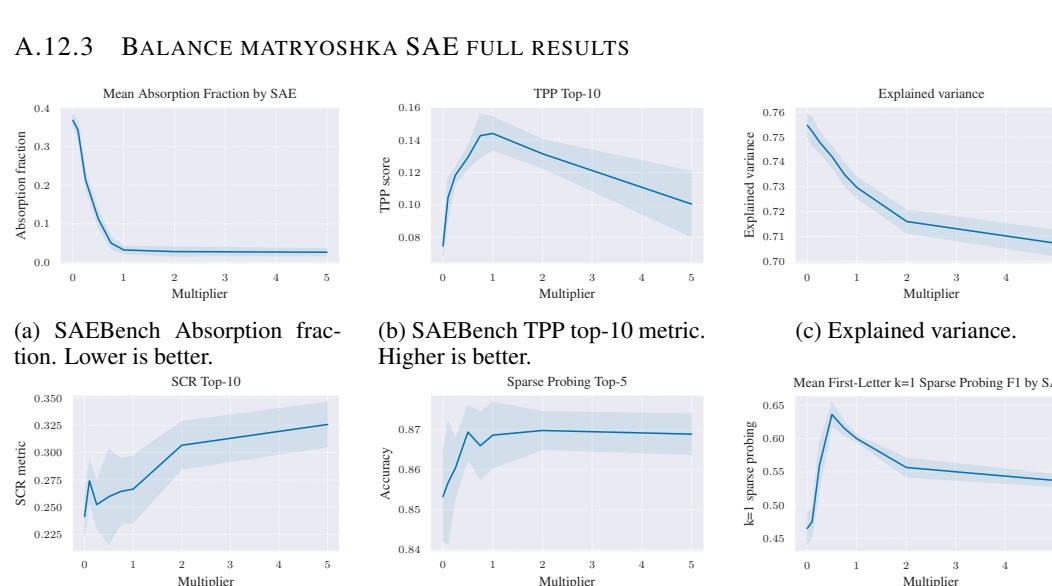

(a) SAEBench Absorption fraction. Lower is better.

(b) SAEBench TPP top-10 metric. Higher is better.

(c) Explained variance.

(d) SAEBench SCR top-10 metric. Higher is better.

(e) SAEBench K=5 sparse probing accuracy.

(f) K=1 first-letter sparse probing F1 score (SAEBench absorption).

Figure 21: Performance of balance matryoshka SAEs vs multiplier for extended metrics. The shaded area in the plots refers to 1 std. Multiplier=0 is equivalent to a standard non-matryoska SAE, and multiplier=1 is equivalent to a standard matryoshka SAE.

### A.13 JUSTIFICATION FOR LEARNING PARENT LATENTS BEFORE CHILD LATENTS

We justify the assumption that an SAE trained with MSE loss will prioritize learning "parent" features over "child" features in a hierarchical relationship. Let $f_p$ be a parent feature and $f_c$ be a child feature such that $f_c \implies f_p$. We assume features are binary, firing with magnitude 1 when active.

The probability of the parent firing is strictly greater than or equal to the child:

$$P(f_p = 1) = P(f_c = 1) + P(f_p = 1 \wedge f_c = 0) \geq P(f_c = 1) \tag{24}$$

Consider an SAE with a single latent $l$ trained to reconstruct a signal $x$ composed of these features. The MSE loss $\mathcal{L}$ for reconstructing a feature $f$ with probability $p$ using a latent with weight $w$ is minimized when $w \approx 1$, yielding a reduction in expected error proportional to $p$.

Specifically, the expected reduction in MSE from perfectly representing feature $f_i$ versus representing zero is:

$$\Delta\mathbb{E}[\mathcal{L}]_i = \mathbb{E}[\|x\|^2] - \mathbb{E}[\|x - f_i\|^2] = \mathbb{E}[2x \cdot f_i - \|f_i\|^2] \tag{25}$$

Assuming unit norm features and orthogonal interference, this simplifies to being proportional to the firing probability $P(f_i)$.

Since $P(f_p) \geq P(f_c)$, the gradient descent process on the MSE objective will experience a stronger driving force to align the latent $l$ with $f_p$ than with $f_c$ in the early stages of training (or in the capacity-constrained initial layers of a Matryoshka SAE). Capturing the parent feature $f_p$ explains more variance in the data than capturing $f_c$, making it the global minimum for a highly constrained (narrow) dictionary.

## A.14 LIMITATIONS

We only test hedging in SAEs up to 65k latents on LLMs with 2b parameters due to compute constraints. Our method for detecting hedging requires fine-tuning SAEs, which is expensive.

