# OpenReview forum: "Feature Hedging: Correlated Features Break Narrow Sparse Autoencoders"
_ICLR.cc/2026/Conference — Submitted to ICLR 2026_

### Official Review · Reviewer_WZV8 · 2025-10-31

**Soundness:** 2
**Presentation:** 3
**Contribution:** 2
**Rating:** 4
**Confidence:** 4

**Summary:**

In this work, the authors have defined and studied the feature hedging problem, theoretically in toy models and empirically in LLM SAEs. We show that hedging is worse the more narrow the SAE, and introduce a technique to characterise the amount of hedging present in a given SAE. They also studied hedging and absorption in matryoshka SAEs. They demonstrated that it is possible to improve the monosemanticity of matryoshka SAEs by adjusting the relative loss coefficients at each level of the matryoshka SAE better to balance the competing forces of absorption and hedging, while both issues remain present. It is also shown that the SAE width is not a neutral hyperparameter: narrow SAEs suffer more from hedging than wider SAEs.

**Strengths:**

Well written, good diagrams, the problem is clearly formulated
The methodology for the hedging degree is well explained
Feature hedging may not be a fully novel idea, buta  good and useful idea for SAE

**Weaknesses:**

Formal proofs are limited; relies mostly on empirical evidence and illustrative derivations.
Some assumptions—e.g., “parent latents are learned before child latents”—while intuitive, are not rigorously tested.
The broad idea that correlation can distort learned dictionaries has appeared in sparse coding literature so not so novel idea
Characterisation of Hedging is missing
\beta multiplier nonbelensing is not clear

**Questions:**

Give more rigorous theoretical proof
explain and reason out assumptions
Potential interactions between reconstruction loss and latent geometry could be explored more formally.
More experimentation can be performed

---

> ### Author Response · Authors · 2025-11-22
>
> We thank the reviewer for engaging with our work and providing feedback. We address the concerns raised below:
>
> > Formal proofs are limited; relies mostly on empirical evidence and illustrative derivations.
>
> We have addressed this by adding a formal theoretical treatment of hedging to Appendix A.4. Specifically:
>
> **Proof of Hedging:** We prove that for an SAE narrower than the ground-truth feature set, if the correlation $\rho$ between a represented feature $f_1$ and an unrepresented feature $f_2$ exceeds a threshold defined by the sparsity penalty, the global minimum of the loss function strictly requires the learned latent to be a polysemantic linear combination of $f_1$ and $f_2$.
>
> **MSE Implications:** We show analytically that "hedging" is not an optimization failure, but the mathematically optimal way to minimize Reconstruction Error (MSE) given a width constraint
>
> > Some assumptions—e.g., “parent latents are learned before child latents”—while intuitive, are not rigorously tested.
>
> We have formalized the logic behind this and added it to Appendix A.13. The logic is as follows:
>
> It is easy to show that learning parent latents before child latents minimizes MSE of the SAE as long as the child latent firing magnitudes are not dramatically higher than the parent latent. In reality, this is a reasonable assumption. The expected MSE contribution of missing feature $f$ is proportional to $P(f)$. By definition, parent latents fire with higher probability than child latents, since every time a child latent fires the parent must also fire. Thus, minimizing MSE should require learning parent features before child features.
>
> Furthermore, this is the core assumption underlying Matryoshka SAEs. If this is false, Matryoshka SAEs should not work. Currently, Matryoshka SAEs achieve the highest scores of any known architecture on multiple SAEBench metrics, further validating this assumption is true.
>
> > The broad idea that correlation can distort learned dictionaries has appeared in sparse coding literature
>
> We added more references to related work to address this concern. While we are aware of work that shows strong coherence between atoms in dictionary learning can break identifiability, all work we are aware of assumes the generating and learned dictionaries are the same size [3, 4]. Our work deal with dictionaries that are *more narrow* than the number of true features.
>
> Recent work shows that SAEs underperform other interpretability techniques [1,2], yet we have not seen any work explaining why this could be. If we do not understand why SAEs are underperforming we cannot find solutions to improve their performance. Our work is thus valuable as it illuminates a core reason that may explain why current SAEs may not be living up the expectations of the field.
>
> > Characterisation of Hedging / more experimentation can be performed
>
> We interpreted this as a request to characterize under what conditions hedging occurs. We have added a large-scale sweep on toy models with controlled correlations in Appendix A.5.
>
> We demonstrate a smooth phase transition: our hedging metric $h$ scales linearly with the degree of feature correlation and inversely with SAE width. Crucially, the metric $h$ drops to zero exactly when the SAE width matches the number of ground-truth features, validating that our metric correctly characterizes the phenomenon.
>
> > Give more rigorous theoretical proof
>
> We have added a proof to Appendix A.4
>
> > explain and reason out assumptions
>
> We have done our best to explain our reasoning and assumptions throughout the paper, building from our theoretical understanding of hedging that we see in toy models (and absorption from previous work) in Section 3 to our hedging metric in Section 4 and balance matryoshka SAEs in Section 5. We have added further proofs and theory in Appendix A.4 and A.13, and moved Section 3.5, demonstrating that MSE is minimized by mixing features, back into the main body.
>
> > \beta multiplier nonbelensing is not clear
>
> We assume “nonbelensing” is referring to cases where perfect balancing is impossible. Appendix A.11 discusses the situation where perfect balancing is impossible using toy model experiments. We demonstrate that even in cases where perfect balancing is not possible, balancing is still a net improvement.
>
> ### References
> - [1] Kantamneni, Subhash, et al. "Are sparse autoencoders useful? a case study in sparse probing." arXiv preprint arXiv:2502.16681 (2025).
> - [2] Wu, Zhengxuan, et al. "Axbench: Steering llms? even simple baselines outperform sparse autoencoders." arXiv preprint arXiv:2501.17148 (2025).
> - [3] Gribonval, Rémi, and Karin Schnass. "Dictionary identification—sparse matrix-factorization via l1-minimization." IEEE Transactions on Information Theory 56.7 (2010): 3523-3539.
> - [4] Hu, Jingzhou, and Kejun Huang. "Global Identifiability of l1-based Dictionary Learning via Matrix Volume Optimization." Advances in Neural Information Processing Systems 36 (2023): 36165-36186.

---

### Official Review · Reviewer_G273 · 2025-10-31

**Soundness:** 2
**Presentation:** 3
**Contribution:** 3
**Rating:** 4
**Confidence:** 3

**Summary:**

This paper describes the phenomenon of feature hedging, in some ways the opposite of the previously noted phenomena of feature absorption. Hedging is analyzed in four toy scenarios: independent, hierarchical, correlated, and anti-correlated features. Then, a metric, hedging degree, is proposed and used to measure hedging in SAEs for LLMs. Finally, the paper proposes balance matryoshka SAEs that scale the loss for each level to reduce hedging. This appears to be a simple approach that can help, although, as noted by the authors, not in all cases.

I learn towards rejecting the paper, mainly because feature hedging is not formally defined. This makes it difficult to judge whether the hedging degree is a reasonable metric, and limits formal understanding of the phenomenon.

Finally, I would like to disclose that I am not familiar with recent work on SAEs beyond the most well-known papers. There has been a surge of works in this area, so there could be closely related works to this paper that I am not aware of.

**Strengths:**

1. **Identifies a fundamental issue**. The paper identifies a fundamental issue in SAEs. I agree with the authors' conclusion in section 7 that understanding hedging is valuable for further work on SAEs.
2. **Balance matryoshka SAE**. This generalization of matryoshka SAEs is appealing. Weighing the levels is a small, simple change, and we can retrieve both SAEs and matryoshka SAEs as special cases, and the paper demonstrates that it can reduce hedging.

**Weaknesses:**

1. **Theoretical analysis**. Although the simple toy examples are useful for understanding when feature hedging may occur, some theoretical analysis of the phenomenon would be a useful complement. Would it be possible to explain why these feature-hedging optima occur in simple toy settings?
2. **Causes of hedging**. Throughout the paper, different causes for hedging are discussed. Some are related to the optimization problem itself (MSE, use of L1), others to the data (feature correlations), and even the learning process (in the paragraph on line 287). My impression is that the actual cause is not fully clear, and the discussion ends up slightly confusing. Again, some theoretical analysis could help clear things up.
3. **Hedging degree**. It is hard to tell whether this metric measures hedging, especially as hedging has not been formally defined. Moreover, it seems like a difficult metric to use in practice, as it depends on a hyperparameter N and training the network (and thus the training setup, such as the optimizer and length of extended training). A.5 states that the hedging degree increases with N, but does not provide a convincing reason for choosing some N. If there was, e.g., a formal definition of hedging and hedging degree was derived as an approximation of it, I would find its use more convincing, but as it stands, it does not appear to be a reliable metric.

**Questions:**

1. **Why not use more latents?**. Perhaps a naive question: why should we not simply make the model wider and focus on reducing feature absorption? If SAEs are "almost certainly narrower than the number of underlying features" (line 72), it would be natural to first make them less narrow, which would reduce hedging?
2. **$\beta$ multipliers**. On line 418, the $\beta$ parameters are parameterized relative to each other using a multiplier. How did you arrive at this particular form? That is: (ii) why is it decreasing, and (ii) why a constant multiplier and not some other function?

**Minor comments**
- Throughout the text, it would be good to disambiguate input features and latent features in the text so that they are not confused (as is done with $f$ and $l$ in mathematical notation).
- Line 17, "is caused by SAE reconstruction loss" -> "is caused by *the SAE's* reconstruction loss".

---

> ### Author Response · Authors · 2025-11-22
>
> We thank the reviewer for engaging with our work and providing thoughtful feedback. We respond to questions and concerns below:
>
> > some theoretical analysis of the phenomenon would be a useful complement
>
> We analytically calculated loss curves for a simple single-latent SAE operating on a 2-feature toy model, showing explicitly where loss is minimized as a function of how much the underlying features are mixed together. This was previously in the Appendix, but we now moved this analysis to the main body Sec 3.5, and added a proof in Appendix A.4.
>
> > different causes for hedging are discussed. Some are related to the optimization problem itself (MSE, use of L1), others to the data (feature correlations) … the cause is not fully clear
>
> The cause is clear. Hedging occurs under the following conditions:
>
> - SAE width is less than the number of underlying true features
> - The underlying true features have correlations between them
>
> If this is the case, then MSE loss is minimized when SAE latents mix underlying features rather than disentangling them.
>
> The confusion around L1 is that the L1 penalty can partially counteract this only in the cause of positive correlations, only when those correlations are not hierarchical, and only if the L1 penalty is very strong. We agree this is confusing, but in reality, we expect that L1 SAEs still suffer significantly from hedging, and we do indeed see that in our LLM experiments.
>
> We hope the newly added single-latent analysis and proof in A.4 should clear this up.
>
> > hedging metric depends on a hyperparameter N and training the network … difficult to use in practice
>
> Our goal with this metric is simply to show that hedging happens in real LLMs, not only toy models, and to allow us to gauge relative amounts of hedging. Since we do not have ground-truth features in real LLMs, we need some way to tell if extending an SAE is removing correlated features from existing latents, and this is the best proxy we could come up with. We agree this metric is difficult to use in practice.
>
> For this purpose, any choice of N that’s used consistently should work. We are trying to invalidate the null hypothesis, which is that there is no hedging, so adding latents to an SAE should have no effect on existing latents beyond random perturbations. Our metric directly measures the extent to which adding latents removes components of the newly added latents from existing latents. This can only happen if hedging occurs in real LLMs.
>
> > why not simply make the model wider and focus on reducing feature absorption?
>
> We very much support this direction! If not for feature absorption, our work would suggest training as wide an SAE as possible will result in a much better SAE. However, we do not know of any way to reduce absorption in wide SAEs aside from Matryoshka SAEs currently. If our work spurs the community to find ways of solving absorption without exacerbating hedging, then we view that as a success.
>
> > $\beta$ parameters are parameterized relative to each other using a multiplier. How did you arrive at this particular form? That is: (ii) why is it decreasing, and (ii) why a constant multiplier and not some other function?
>
> We only have limited compute and cannot do an exhaustive sweep of every possible way of setting the $\beta$, and thus, we had to pick some reasonable way to conduct a sweep. We would try other methods if we had the compute for it, and in-fact hope to further develop heuristics to optimize all Matryoshka hyperparameters in the future.

---

### Official Review · Reviewer_3uzz · 2025-11-04

**Soundness:** 2
**Presentation:** 3
**Contribution:** 4
**Rating:** 4
**Confidence:** 4

**Summary:**

The paper introduces the problem of "feature hedging" in Sparse AutoEncoders (SAEs), motivating it first in the context of "toy" models, then empirically studying it in the context of larger-scale SAEs/LLMs. The authors proceed to introduce a tentative solution to hedging by incorporating a balancing term in the Matryoshka SAE loss function, finding that doing so presents empirical benefits for "Balance Matryoshka SAEs" relative to unbalanced baselines.

**Strengths:**

- The authors motivate, define, empirically study, and propose tentative solutions to a new and potentially important problem ("feature hedging") with leading interpretability methods (SAEs).
    - Both defining/demonstrating this problem, and making substantive empirical progress towards understanding and resolving it, represent significant contributions to the interpretability community.
- The paper is generally well-written, with clear argumentation and strong intuition-building in both the introduction and "toy" experiments (sec 3).
- Hedging experiments (sec 4) are comprehensive, considering a range of LLMs/layers, SAE architectures and hyperparameters, etc.
    - And reporting several categories of SAEBench scores across hyperparameter values for the balancing multiplier (in sec 5) is also much appreciated for better understanding and contextualizing balancing with respect to standard (multiplier = 0) or unbalanced Matryoshka (multiplier = 1) SAEs.

**Weaknesses:**

The most substantial weakness is the lack of hedging results for Balance Matryoshka SAEs (elaborated below) -- given that the core motivation for the proposed balancing approach is to resolve the hedging problem, the absence of results showing whether balancing actually helps at all with this problem means that it cannot be taken seriously as a contribution. There are also more minor concerns regarding clarity on some important experimental details (discussed in the Questions section of the review) and citing prior interpretability work only within the narrow context of SAEs (elaborated below). Weaknesses by section are provided below.

Background/Related Work:
- There is already substantial work on closely-related problems in the context of probing classifiers -- see, e.g., [1-5]. (Note that this is not a serious novelty concern, as hedging is an SAE-specific problem and the most relevant works studying SAEs are already cited and discussed. Instead, I believe this is simply another instance of the "parallel community" studying mechanistic interpretability largely focused on SAEs that often fails to cite work from the broader, longstanding interpretability/model analysis literature, as highlighted in [6].)

[1] Kumar, A., Tan, C., & Sharma, A. (2022). Probing classifiers are unreliable for concept removal and detection. Advances in Neural Information Processing Systems, 35, 17994-18008.
[2] Ravichander, A., Belinkov, Y., & Hovy, E. (2021, April). Probing the Probing Paradigm: Does Probing Accuracy Entail Task Relevance?. In Proceedings of the 16th Conference of the European Chapter of the Association for Computational Linguistics: Main Volume (pp. 3363-3377).
[3] Elazar, Y., Ravfogel, S., Jacovi, A., & Goldberg, Y. (2021). Amnesic probing: Behavioral explanation with amnesic counterfactuals. Transactions of the Association for Computational Linguistics, 9, 160-175.
[4] Canby, M., Davies, A., Rastogi, C., & Hockenmaier, J. (2024). How Reliable are Causal Probing Interventions?. arXiv preprint arXiv:2408.15510.
[5] Hewitt, J., & Liang, P. (2019). Designing and interpreting probes with control tasks. arXiv preprint arXiv:1909.03368.
[6] Saphra, Naomi, and Sarah Wiegreffe. "Mechanistic?." Proceedings of the 7th BlackboxNLP Workshop: Analyzing and Interpreting Neural Networks for NLP. 2024.

Sections 4-5:
- In sec 5, it is claimed that "As we saw in Section 4.1, the more narrow an SAE is, the worse the hedging. Matryoshka SAEs thus solve feature absorption at the expense of exacerbating feature hedging." However, the experiments in sec 4.1 do not explicitly consider Matryoshka SAEs, weakening the motivation for Balance Matryoshka SAEs in sec 5. I suggest repeating the experiments in sec 4.1 (or a suitable subset, given compute constraints) with Matryoshka SAEs to determine the extent to which feature hedging is actually an issue for baseline Matryoshka SAEs.

Section 5:
- **(MOST IMPORTANT:)** There is no measurement of hedging for "real-world" balance Matryoshka SAEs (i.e., those trained on actual LLMs rather than toy models), so it is impossible to tell how well balancing actually solves this problem (relative to baseline Matryoshka SAEs without balancing). The results from "toy" models are useful for illustrating the tradeoff between hedging and absorption, but insufficient to claim that Balance Matryoshka SAEs present any real utility for resolving hedging in the target context of LLMs. I suggest repeating the experiments from sec 4.1 (or a suitable subset, given compute constraints) with the Balance Matryoshka SAEs trained in sec 5.

If the authors are able to perform the experiments noted above (perhaps at a smaller scale if this is infeasible during the discussion period) -- and if results show that Balance Matryoshka SAEs (at some multiplier between 0 and 1) do actually, nontrivially improve on baseline traditional SAEs (multiplier = 0) and Matryoshka SAEs (multiplier = 1) -- then I would be happy to increase my score.

**Questions:**

Intro:
- In the caption of fig 1b, the authors state that "Asymmetry between encoder and decoder is characteristic of absorption", which seems to be critical to the discussion in sec 4 (lines 291-296), and is also reiterated in tab 1. Can you elaborate on this point? Is this expected *a priori* (and in which case, what is the theoretical justification); or is it based on empirical findings in this or other work (in which case, can you cite the relevant work/section)?

Sec 3:
- In the paragraph "The implications of this for SAE performance are quite dire [...]" -- intuitively, I would tend to agree with that this is a problem, but I think the argument could be better fleshed out. For instance, what would the consequences be for practical interpretability desiderata, such as OOD detection or safety monitoring? Why is negative feature mixing an issue if two features really are strongly negatively correlated?
    - The simple answer to the second question is that such correlations may be spurious and lead to poor OOD robustness of SAE features; but for SAE training datasets that are i.i.d. wrt model training data, such issues might still be indicative of the underlying representation learned by models, and thus useful for detecting spurious/shortcut feature learning. I am curious to hear how the authors would respond to this argument, and the degree to which it presents a real challenge to the core motivation of this work.

Sec 4:
- A central point supporting the experiments and interpretation throughout sec 4 is stated as: "Based on our understanding of hedging in toy models, we expect that when a new latent is added to an SAE, this should 'pull out' the component of the new feature from existing SAE latents, where it was previously hedged. Thus if hedging occurs, the change in existing latents after a new latent is added should project onto that new latent." I have a few questions on this point:
    - To my understanding, the evidence in sec 3 and app A.1-2 comes from experiments where the number of latents is fixed prior to training the model -- which is a completely different setting from the expectation stated above. So:
        - Is my understanding here correct?
        - And if so, then what is the stated expectation based on? (I.e., please articulate precisely what it is in your "understanding of hedging in toy models" that would lead to this expectation, and why.)
    - Additionally, I feel that this argument would substantially benefit from a more rigorous formal presentation. For instance: (a) how could one mathematically state the expectation as a hypothesis to be tested, (b) how would one define a reasonable corresponding "null hypothesis" and measure the degree to which one is to be favored over the other, and (c) how could one formally state the evidentiary basis (per the previous questions) in relation to both hypotheses? (Note: I believe that (a) may already be covered by equation 7 and supporting text, but I would appreciate more clarification on precisely how it relates to the stated expectation/hypothesis; and I don't see anything corresponding to (b-c) in the main paper.)
- Similar to the previous question, what would one expect for a "null hypothesis" value of hedging degree h? (A control condition here would be useful -- if this is nontrivial to define in the context of LLM SAEs, then at least showing what h looks like for "toy" experiments like those in sec 3 (ideally across multiple dictionary sizes, k/L0 values, etc, at values closer to those in sec 4) would be helpful.)
- In fig 6,
    - In legends of all 3 plots, am I correct in assuming that "btk" refers to BatchTopK and non-btk is "l1" (meaning L1-regularized SAEs, such as those in Olah et al., 2024)? This was confusing to figure out.
    - What is being visualized for BatchTopK models on on the x-axis of fig 6c? To my understanding, the same k for BatchTopK is used for training all SAEs (with k=25, per appendix A.4) -- so how can there be multiple L0 values? Or are these separate models trained with different values of k?

Sec 5:
- The experiment whose results are reprted in fig 7 needs some clarification -- e.g., even after consulting the appendix, the following are still unclear:
    - What are the dictionary sizes $\mathcal{M} = m_1, ..., m_n$? My best guess is that there are just two dictionaries, $m_1, m_2 = 1, 4$, but this needs to be clarified.
    - $\beta = 0.25$ shows the best results. Does this mean that $\beta_{m_i} = 0.25$ for all $m_i$, or just $m_i : i > 1$?
- What is the basis for the multiplier formula used to obtain $\beta_m$ in lines 415-419? (I understand that the multiplier, such as 0.5, is a hyperparameter; but why set $\beta_m = \mu^{(n - m)}$ for multiplier $\mu$ rather than, e.g., sampling linearly between 0 and 1?)

Minor clarifications/nitpicks:
- In fig 6, the caption states "Hedging degree for SAEs trained on Gemma-2-2b layer 12" -- but both gemma and llama are tested, and fig 6b is over multiple layers, correct? Please update the caption accordingly.
- In fig 8, what is the purpose of setting the multiplier to a value greater than 1? I don't understand what this would correspond to theoretically; and empirically it seems that most variation occurs within [0, 1] (as one would intuitively expect).

---

> ### Author Response · Authors · 2025-11-22
>
> We thank the reviewer for their thorough engagement with our work and for providing so much feedback. We respond to the points raised below, but it will require several comments to address all points raised (apologies for the long replies):
>
> > lack of hedging results for Balance Matryoshka SAEs
>
> This is a limitation of our hedging metric, which makes it hard to detect hedging inside the inner levels of Matryoshka SAEs. The problem is we need a way to extend just an inner Matryoshka level, but this should pull latents from the next level into the inner level, and we could not find a principled way to do this without breaking the outer levels of the SAE.
>
> Fortunately, though, real-world metrics are what really matters, and we can and do evaluate balance Matryoshka SAEs on SAEBench. We use the theory and results we develop throughout the paper to develop and motivate the balance matryoshka architecture, and then show that optimizing the $\beta$s of balance Matryoshka SAEs improves these metrics in Figure 9.
>
> > … closely-related problems in the context of probing classifiers
>
> We are happy to cite work from related fields. However after reading through the listed papers, we struggle to see the connection to feature hedging. The issue we are exploring is the effect of feature correlation on SAEs that are narrower than the number of underlying model features. We did not see probing papers dealing with this problem, as this is not a problem that supervised probes will encounter (if we understand the referenced papers correctly). If you can explain the connection and relevance, we will be happy to add these to related works.
>
> > Asymmetry between encoder and decoder is characteristic of absorption
>
> This is from the original absorption work [1], where they show this both empirically in LLMs and toy models and also have a proof. We replicate their toy model results in Figure 1 as well.
>
> > what would the consequences be for … OOD detection or safety monitoring? Why is negative feature mixing an issue?
>
> To put this in the context of probing, this would be like training thousands of probes for different concepts, and then randomly mixing positive and negative components of each probe into each other probe. If we did such a thing, surely the resulting probes will be dramatically worse in all aspects. This is roughly what we expect to happen from feature hedging into SAE latents. This can only harm performance, as we are just corrupting every learned feature.
>
> Negative feature mixing is a particularly bad issue because there is no reason to expect that the negative of a feature direction has any meaning at all in the model, or even that it can arise naturally. Thus, if we use the SAE latent for steering, the direction we are steering with is likely completely OOD for the model. There is never a case where negative components of every anti-correlated feature can be present at the same time in the same activation, so we should expect this to break steering too.
>
> > but for SAE training datasets that are i.i.d. wrt model training data, such issues might still be indicative of the underlying representation learned by models
>
> We view the goal of SAEs as uncovering the real, underlying representation used by models. If the model itself is representing features in strange ways that are hard for humans to understand, we want to be able to trust that our interpretability tools will still recover those, and will not instead merge together mixes of features simply due to hedging. Our experiments show that currently SAEs will abuse correlations between features when the SAE is too narrow, corrupting the SAE dictionary, so we cannot trust that the features learned are accurate representations of the model’s own representations.

---

> ### Author Response · Authors · 2025-11-22
>
> > the evidence in sec 3 and app A.1-2 comes from experiments where the number of latents is fixed …
>
> > (a) how could one mathematically state the expectation … (b) how would one define a "null hypothesis" … (c) … formally state the evidentiary basis in relation to both?
>
> We added new experiments in Appendix A.4 on large toy models with correlated features where we extend SAEs and calculate our hedging metric, showing that, as we expect:
> - The hedging metric is higher the narrower the SAE.
> - The hedging metric is zero when the SAE width matches the number of true features in the model.
>
> The experiments in S.3 (and A.1-2) are showing that when the SAE is more narrow than the underlying number of features, the SAE will merge components of correlated features that the SAE is not wide enough to represent into the latents of the SAE. The reason we show an SAE that’s too narrow side-by-side with an SAE that’s wide enough trained on the same data distribution is so readers can build intuition around what adding a new latent to the SAE engaged in hedging will do. For each of these cases, you can mentally subtract latents 1-3 on the left from latents 1-3 on the right, and see what the “delta” is between these two SAEs. Specifically, in all cases, the delta exists only in latent 3 and projects only onto latent 4 in the right-hand side. This is the direct motivation for the hedging metric.
>
> The null hypothesis is simply that adding a new latent should have no predictable effect on existing latents. If our hypothesis is incorrect, and the SAE being too narrow does not cause any mixing of features outside the SAE into latents in the SAE, then extending the width of an SAE should not cause any change to existing latents that projects onto newly added latents beyond what might be expected from random chance.
>
> We need to do this because we do not have ground-truth data in real LLMs, so to prove that hedging is happening, we need to use the deltas between SAE widths to infer its presence, which is what our metric does. Below, we formalize this mathematically:
>
> **(a) Hypothesis (**$H_1$**, Hedging):**
>
> Let $S_0$ be an SAE with decoder $W^0_{\text{dec}}$ that is narrower than the set of true features $F$. Due to correlation between a tracked feature $f_i$ and an untracked feature $f_{\text{missing}}$, a latent $l_k$ in $W^0_{\text{dec}}$ learns a mixed representation to minimize MSE:
>
> $$W^0_{\text{dec}}[k] \approx f_i + \gamma f_{\text{missing}}$$
>
> where $\gamma \neq 0$ represents the hedging component.
>
> When we extend the SAE to $S_1$ by adding $N$ new latents ($W^1_{\text{new}}$), and assuming the SAE optimization finds the optimal solution where a new latent captures the previously missing feature ($W^1_{\text{new}}[j] \approx f_{\text{missing}}$), the original latent $l_k$ no longer needs to hedge. It converges to the pure feature:
>
> $$W^1_{\text{dec}}[k] \approx f_i$$
>
> The change vector $\delta_k$ is therefore:
>
> $$\delta_k = W^1_{\text{dec}}[k] - W^0_{\text{dec}}[k] \approx -\gamma f_{\text{missing}}$$
>
> **Expectation:** Since $W^1_{\text{new}}[j] \approx f_{\text{missing}}$, the change vector $\delta_k$ will have a significant non-zero projection onto the subspace spanned by the new latents $W^1_{\text{new}}$.
>
> **(b) Null Hypothesis (**$H_0$**, Random Noise):**
>
> The change in existing latents $\delta_k$ is driven by stochasticity in the optimization process (random seed variation, data shuffling) or re-balancing of uncorrelated noise, unrelated to the specific semantic content of the new latents.
>
> **Expectation:** $\delta_k$ will be a random vector in the high-dimensional activation space $\mathbb{R}^D$. In high-dimensional spaces, a random vector $\delta_k$ is expected to be nearly orthogonal to the specific subspace spanned by $W^1_{\text{new}}$.
>
> **(c) Evidentiary Basis (The Metric** $h$**):**
>
> We define our hedging degree $h$ (Eq. 7) as the difference between the projection under $H_1$ and the projection under $H_0$.
>
> $$h = \mathbb{E}[\text{Projection onto } W^1_{\text{new}}] - \mathbb{E}[\text{Projection onto } W_{\text{rand}}]$$
>
> Here, $W_{\text{rand}}$ serves as the empirical control for the null hypothesis (random directions).
>
> - If $h \approx 0$, we fail to reject $H_0$; the changes in latents are indistinguishable from random noise relative to the new directions.
>
> - If $h > 0$, we reject $H_0$ in favor of $H_1$; the existing latents are systematically shifting _parallel_ to the newly learned features, confirming that the component now captured by the new latents was previously present (hedged) in the old latents.

---

> ### Author Response · Authors · 2025-11-22
>
> > In legends of all 3 plots, am I correct "btk" refers to BatchTopK and non-btk is "l1"
>
> Yes, exactly. We have updated the manuscript to clarify this.
>
> > What is being visualized for BatchTopK models on the x-axis of fig 6c? … are these separate models trained with different values of k?
>
> These are separate SAEs trained with different K. Our goal was to check if the amount of hedging due to width changes for SAEs with different L0. We have clarified this in the caption.
>
> > In fig 6, caption …
>
> Thank you for pointing this out, the caption has been updated.
>
> > In fig 8, what is the purpose of setting the multiplier to a value greater than 1? I don't understand what this would correspond to theoretically; and empirically it seems that most variation occurs within [0, 1] (as one would intuitively expect).
>
> A multiplier of 1 is completely arbitrary, and there is no reason to expect this is an optimal choice in either direction. Empirically we see less than 1 is optimal, but there is no theoretical reason why this should be the case. If hedging were not a problem, we would expect that detaching gradients entirely between Matryoshka levels (equivalent to each $\beta = \infty$) would be ideal, as we would want no interference at all from outer Matryoshka levels on inner levels. In fact, the original Matryoshka SAEs work mentions they tried detaching gradients between levels but found it worked less well for reasons they didn’t understand [2] (see their Appendix G.2) - we now know that reason is hedging. In fact this is already a very strong signal that hedging is a serious issue in LLM SAES, since there is no other theoretical reason we know of that detaching matryoshka levels will result in a worse SAE aside from feature hedging.
>
> ### References
> - [1] Chanin, David, et al. "A is for absorption: Studying feature splitting and absorption in sparse autoencoders." arXiv preprint arXiv:2409.14507 (2024).
> - [2] Bussmann, Bart, et al. "Learning multi-level features with matryoshka sparse autoencoders." arXiv preprint arXiv:2503.17547 (2025).**

---

### Official Review · Reviewer_esmW · 2025-11-07

**Soundness:** 2
**Presentation:** 1
**Contribution:** 2
**Rating:** 2
**Confidence:** 3

**Summary:**

The paper studies LLM interpretability with sparse auto-encoders (SAEs). Previous works identify one phenomenon where one feature actives only when another does as the limiting factor for applying SAEs for interpretability. The authors introduce another limiting phenomenon which they call "feature hedging", that occurs when the SAE's latent space is too narrow and the underlying features are correlated.

They demonstrate this through some toy experiments on 2 and 4 dimensional settings and they define a metric to detect this effect in LLMs. Their experiments on SAEs trained on LLM activations, and report their results using the metric (which they call hedging degree) as a function of the parameters of SAE and average number of active latents. Finally they propose a re-weighted Matryoshka training to overcome the phenomenon they proposed.

**Strengths:**

The idea of studying sparse autoencoders under the assumption of anti-correlated (or hierarchical) features is a good direction, the toy setup the authors propose is interesting and intuitive, and in the tests they show using LLMs, the hedging-degree metric appears to work.

**Weaknesses:**

While the direction of study is novel, the large scale experiments are limited in scope, and I am not fully convinced that the metric of hedging degree is the right one to measure this phenomenon. Is the $\lVert .\rVert$ an $\ell_2$ or $\ell_1$ norm? Further, what is $W_{rand}[0:N]$? are the latents initialized with a Uniform or a Gaussian distribution?

Can the authors give more reasoning behind selecting this metric besides it exceeding a "random" baseline? It would help their case to see tests using this metric on the toy setups.

Overall, the writing can be greatly improved to a more precise way of defining terms. As an example, terms like polysemantic and monsemantic activations without their definitions, making this paper sort of inaccessible to users outside the area of interpretability.

Overall, I do not recommend accepting this paper.

**Questions:**

Asked above.

---

> ### Author Response · Authors · 2025-11-22
>
> We thank the reviewer for their reading of our work and feedback. We answer questions and concerns below:
>
> >  Is the $\lVert .\rVert$ an $\ell_2$ or $\ell_1$ norm?
>
> This is L2 norm. We have added $_2$ to clarify this in the manuscript.
>
> > Further, what is $W_{rand}[0:N]$? are the latents initialized with a Uniform or a Gaussian distribution?
>
> This is described in the manuscript S.4 as follows: $W_\text{rand}[0:N]$ refers to a decoder consisting of $N$ randomly initialized unit-norm latents.
>
> Because all latents are unit-norm there is no difference between a zero-centered uniform or gaussian initialization in this case - the result is that every possible vector in the space has equal probability. In practice we use a gaussian distribution, but using a uniform distribution is identical as long as it has zero centered.
>
> > Can the authors give more reasoning behind selecting this metric besides it exceeding a "random" baseline? It would help their case to see tests using this metric on the toy setups.
>
> We have now added larger-scale toy model experiments with correlated features to Appendix A.5 verifying that the hedging metric works exactly as predicted:
>
> - The more narrow the SAE, the higher the hedging degree
> - When the SAE width matches the number of features in the underlying model, the hedging metric is zero
>
> We describe the rationale behind this metric in Section 4. We summarize that rationale here:
>
> The hedging metric $h$ quantifies the extent to which adding capacity (new latents $W_{new}$) allows existing latents $W_{old}$ to "unlearn" components of features they were previously hedging. If hedging occurs, an existing latent $l_k$ is a mix of a tracked feature $f_i$ and an untracked, correlated feature $f_{missing}$. When new latents are added that capture $f_{missing}$, $l_k$ should converge to the pure $f_i$. This change $\delta_k$ in the existing latent will therefore have a component parallel to the new latent $f_{missing}$. The metric $h$ measures the projection of this change $\delta_k$ onto the subspace spanned by the new latents, subtracting a baseline projection onto random directions to account for noise. A positive $h$ indicates systematic unlearning of hedged features.
>
> > terms like polysemantic and monsemantic activations without their definitions, making this paper sort of inaccessible to users outside the area of interpretability
>
> These terms are defined explicitly and formally in the manuscript on L162-172. If this definition is not sufficient let us know why and we can change it. This text is copied verbatim below for convenience:
>
> We say that an SAE is *correct* or *monosemantic* for this toy model if every latent in the SAE dictionary matches a true feature direction, and each SAE latent corresponds to a different true feature. Formally, there exists a bijection $\pi: \{1, \ldots, L\} \to \{1, \ldots, N\}$ such that $\cos(W_{\text{dec},i}, f_{\pi(i)}) = 1$ for all $i \in \{1, \ldots, L\}$. We only investigate SAEs where $L \leq N$ in our toy experiments. We say an SAE is *polysemantic* if some SAE latents contain positive or negative components of multiple true features, so there exists at least one latent $i \in \{1, \ldots, L\}$ such that $|\{j \in \{1, \ldots, N\} : |W_{\text{dec},i} \cdot f_j| > \epsilon\}| > 1$ for some threshold $\epsilon > 0$.

---

### Author Response · Authors · 2025-11-30

We thank the reviewers for their feedback during the review period. We feel this review process has significantly strengthened the theoretical and empirical grounding of our work. Below, we summarize the key concerns raised by reviewers and the improvements made to the paper to address them.



**Theoretical Proof of Feature Hedging (reviewers WZV8, G273)**

Reviewers requested a more rigorous theoretical foundation for feature hedging.

- **Revision:** We added a formal derivation proving that "hedging" is not merely an optimization failure, but the *mathematically optimal strategy* for minimizing MSE when an SAE is narrower than the number of underlying features ($L_{SAE} < N_{features}$) and features are correlated (see Appendix A.4).
- **Revision:** We moved the section on loss curve analysis where we explicitly calculate the minimum MSE loss as a function of feature mixing, from the appendix to the main body (Section 3.5) to better center the theoretical argument.

**Validation of the "Hedging Degree" Metric (reviewers esmW, 3uzz, G273)**

Reviewers requested further toy model experiments to validate the Hedging Degree metric ($h$) and whether it effectively measures the phenomenon.

- **Revision:** We added large-scale experiments on toy models with ground-truth features (Appendix A.5). These experiments demonstrate that the hedging metric $h$ decreases as the SAE is widened, as expected. Crucially, *$h$ drops to exactly zero* when the SAE width matches the number of ground-truth features. This validates $h$ as a reliable proxy for detecting feature mixing in real LLMs where ground truth is unknown.
- **Clarification:** We provided a mathematical formulation of the hypothesis ($H_1$) and null hypothesis ($H_0$) underlying the metric in our response to Reviewer 3uzz to clarify the evidentiary basis of the projection method.

**Justification of Learning Dynamics (reviewer WZV8)**

The reviewer asked for us to formalize the argument regarding the assumption that "parent" latents are learned before "child" latents in hierarchical features.

- **Revision:** We formalized the argument that minimizing MSE implies learning high-probability (parent) features before conditional (child) features (Appendix A.13). This assumption is fundamental to the operation of Matryoshka SAEs and is supported by their state-of-the-art performance on standard benchmarks.

---

### Meta-Review · Area_Chair_fxB5 · 2026-01-18

**Summary:**

The authors are interested in the conditions under which SAEs can decompose independent features from neurons. Their core finding is a new phenomenon they term feature hedging, where individual dimensions in an SAE can correspond to multiple underlying true features. They propose a new metric for detecting this phenomenon and applied it in toy and large scale settings. They use their understanding to

Reviewers raised concerns about the motivation of the metric, and more broadly the theoretical insight that was gained into feature hedging in this work -- it's causes and its consequences for interpretability work broadly.
One reviewer was concerned that although the description of feature hedging nominally motivated a new type of SAE which performed better on interpretability benchmarks, the authors were unable to demonstrate that the reason for this improvement was due to a reduction in feature hedging.

**Reviewer Concerns:**

The authors clarified definitions, and clarified some connections between feature hedging and past work. Although they didn't resolve the theoretical weaknesses or flesh out a connection between their practical and theoretical results.

**Reviewer Scores:**

Reviewer esmW seemed to have challenges understanding the material. Their definitional and motivation questions were answered with references to the paper. Given their feeling about the writing, they may have stayed at a 2 or gone up to a 4.

Reviewer 3uzz stated explicitly in their original review that they would not raise their scores due to a disconnect between the new proposed SAE and the theoretical contributions.

Reviewer G273 had some of their questions answered but their main concerns about theoretical insights remain.

Reviewer WZV8 was not engaged in the review process.

---

### Decision · Program_Chairs · 2026-01-26

Reject